# The effects of a comparatively higher dose of 1000 mg/kg/d of oral L- or D-arginine on the L-arginine metabolic pathways in male Sprague-Dawley rats

**Dain (Raina) Kim, Sarah Martin, Kaushik Desai** [ORCID] *

Department of Anatomy, Physiology & Pharmacology, University of Saskatchewan, Saskatoon, SK, Canada

* k.desai@usask.ca

**Data Availability Statement:** We have uploaded our minimal underlying data set on DRYAD: https://datadryad.org/stash/share/4jH7mMbb8DzurfoCJz

## Abstract

Oral L-arginine supplements are popular mainly for their nitric oxide mediated vasodilation, but their physiological impact is not fully known. L-arginine is a substrate of several enzymes including arginase, nitric oxide synthase, arginine decarboxylase, and arginine: glycine amidinotransferase (AGAT). We have published a study on the physiological impact of oral L- and D-arginine at 500 mg/kg/day for 4 wks in male Sprague-Dawley rats. We investigated the effects of oral L-arginine and D-arginine at a higher dose of 1000 mg/kg/d for a longer treatment duration of 16 wks in 9-week-old male Sprague-Dawley rats. We measured the expression and activity of L-arginine metabolizing enzymes, and levels of their metabolites in the plasma and various organs. L-arginine did not affect the levels of L-arginine and L-lysine in the plasma and various organs. L-arginine decreased arginase protein expression in the upper small intestine, and arginase activity in the plasma. It also decreased AGAT protein expression in the liver, and creatinine levels in the urine. L-arginine altered arginine decarboxylase protein expression in the upper small intestine and liver, with increased total polyamines plasma levels. Endothelial nitric oxide synthase protein was increased with D-arginine, the presumed metabolically inert isomer, but not L-arginine. In conclusion, oral L-arginine and D-arginine at a higher dose and longer treatment duration significantly altered various enzymes and metabolites in the arginine metabolic pathways, which differed from alterations produced by a lower dose shorter duration treatment published earlier. Further studies with differing doses and duration would allow for a better understanding of oral L-arginine uses, and evidence based safe and effective dose range and duration.

## Introduction

L-Arginine (L-Arg) is a semi-essential amino acid that contributes to the production of many biologically important metabolites through multiple enzymatic pathways [1]. D-Arginine (D-Arg) on the other hand is generally known to be the metabolically inert isomer, and therefore is frequently disregarded in oral arginine supplement studies. D-Arg produced some significant effects on L-Arg metabolizing enzymes and their metabolites in our previous study [2]. D-Arg can be used by animals after inversion to the L-isomer [3]. The four main enzymes

MG1A1pjAyirfB1hJPW9AzPCME DOI: https://doi.
org/10.5061/dryad.x95x69pqm.

Funding: KD- Grant # RGPIN-2016-03951, Natural
Sciences and Engineering Council of Canada
(NSERC), Discovery Grant, https://www.nserc-
crsng.gc.ca/index_eng.asp. The funder had no role
in study design, data collection and analysis,
decision to publish, or preparation of the
manuscript. DK - Natural Sciences and Engineering
Council of Canada (NSERC), https://www.nserc-
crsng.gc.ca/index_eng.asp., Alexander Graham Bell
Canada Graduate Scholarship, The funder had no
role in study design, data collection and analysis,
decision to publish, or preparation of the
manuscript.

Competing interests: The authors have declared
that no competing interests exist.

that use L-Arg as a substrate are arginase, nitric oxide synthase (NOS), arginine:glycine amidi-notransferase (AGAT/GATM), and arginine decarboxylase (ADC) [4]. Arginase is a major enzyme that converts L-Arg into L-ornithine and urea, and thereby is known for its important role in the urea cycle, for detoxifying ammonia. The arginase I isoform is localized in the liver for urea formation [5], whereas arginase II is distributed mainly in the kidney, pancreas, and the intestines for mainly polyamine synthesis, through ornithine and ornithine decarboxylase (ODC) [4, 5]. NOS is another major enzyme that facilitates the conversion of L-Arg into nitric oxide (NO) and L-citrulline. NO is a potent vasodilator, and the vasodilatory effect is the primary reason for the wide use of oral L-Arg supplements. Endothelial nitric oxide synthase (eNOS) is mostly expressed in endothelial cells and produces NO in a pulsatile manner to cause vasodilation and prevent platelet aggregation [6]. Arginase and eNOS have been recognized to compete for L-Arg and interfere with the activity of one another [7], which can be problematic as NO deficiency is linked with endothelial dysfunction and cardiovascular diseases [8–10]. ADC metabolizes L-Arg into agmatine, which subsequently gets hydrolyzed into putrescine by agmatinase to form other polyamines [11]. Polyamines play important roles in cell growth and support for embryonic development [11]. It is generally believed that arginase is the primary enzyme and ADC is the alternative enzyme for polyamine synthesis [11].

Finally, AGAT coordinates with guanidinoacetate N-methyltransferase to convert arginine into creatine, which is important for energy regulation in the body [12]. Using Guanido-labeled (C-14) arginine in rats, Frondoza et al. [13] showed that after i.p administration peak plasma levels reached within 30 min and only 0.4 percent remained in total body plasma, and 32 percent appeared in the urine and only 3 percent in the stools after 24 hour. The skin, liver and small intestine retained the highest percent of radioactivity. In another study in humans, using [$^{15}$N-$^{15}$N-(guanidino)]-arginine, Mariotti et al. [14] showed that 60 percent of the dietary arginine was converted to urea in first pass metabolism and that <0.1 percent of the dose contributed to NO formation. The kinetics of metabolism suggested competition between the arginase and NOS metabolic pathways.

Oral L-Arg supplements have been in use by both healthy and people with different disease conditions, mainly for its presumed vasodilatory effects. For instance, L-Arg supplements are used by athletes for ergogenic effects [15], by patients with erectile dysfunction [16, 17], hypertension [18], hypercholesterolemia [19], and type 2 diabetes [20]. Although there is unregulated consumption of oral L-Arg supplements, the safety and efficacy, as well as an evidence based daily dose range and durations for different conditions, of these supplements have not been established. We recently published a study on the physiological impact of L-Arg and D-Arg at a daily dose of 500 mg/kg/d for 4 wks in male Sprague-Dawley rats where we described significant effects on several enzymes and metabolites of L-Arg [2]. In this study we tested the effects of oral L-Arg and D-Arg, each at a higher dose of 1000 mg/kg/d for longer duration of 16 wks. We hypothesized that oral L-Arg and D-Arg treatment, at a higher dose and longer duration, will replicate and magnify the effects of 500 mg/kg/d for 4 wks, and possibly affect more enzymes and metabolites of L-Arg, allowing data collection towards establishing a safe and effective dose range and duration for oral supplements that result in increased NO bioavailability. Our results were different than expected.

## Materials and methods

### Animals

Eighteen 8 wk old male Sprague-Dawley (SD) rats were purchased from Charles River Canada (Montreal, QC, Canada) for use according to a protocol (#20160059) approved by the Animal Care Committee at the University of Saskatchewan, following guidelines of the Canadian Council on Animal Care and the ARRIVE guidelines [21]. The rats were fed standard lab rat

diet (Prolab® RMH 3000, LabDiet, St. Louis, MO, USA) containing L-Arg at 1.37 percent in a protein content of 22.5 percent. This amounts to about 1170 mg/kg/d of dietary L-Arg for a 350 g rat. After one week of acclimatization the rats were randomly assigned to one of the three treatment groups: (1) Control group: plain drinking water (*ad libitum*) ($n = 4$), (2) L-Arg group: L-arginine (1000 mg/kg/d) in drinking water ($n = 7$), (3) D-Arg group: D-arginine (1000 mg/kg/d) in drinking water ($n = 7$). The treatment was provided for 16 wks. For the L-Arg and D-Arg stock solutions, the L-Arg and D-Arg free bases (L-Arg, Cat # W381918, Sigma-Aldrich Canada Ltd; D-Arg, Cat # GM7267, Glentham Life Sciences, Corsham, U.K.) were dissolved in drinking water, and the pH was adjusted to 7.4 with hydrochloric acid. The D-Arg solution had a pale yellowish colour and a faint odour and the D-amino acids are known to have different colour, odour and taste than the L-isomers. Each rat was placed alone in a cage, and the body weights of all the rats were recorded immediately before and after treatment and every other day during the treatment period. As well, the daily water intake for each rat was recorded from its individual cage. To ensure that all rats get the same dose through the drinking water, the dose of L-Arg and D-Arg to be added to the drinking water for each rat was calculated with a specially devised formula: Body weight (g) x 15 / water intake per day (mL) = X mL of 20 g/L stock L-Arg or D-Arg solution to be added to a 300 mL bottle and the rest tap water to make 300 mL. The L-Arg and D-Arg doses for each rat were calculated and adjusted in the drinking water every other day. For example, for a rat weighing 350 g and drinking 60 mL of water per day we would add 87.5 mL of stock (20 g/L) L-Arg or D-Arg solution to a 300 mL bottle, which would make 1750 mg L-Arg or D-Arg in the 300 mL bottle. When the rat drinks 60 mL per day it will receive 350 mg, i.e. 1000 mg/kg/d. Following the 16-wk treatment, the rats in all three groups were individually placed in metabolic cages to collect urine over a 16 hour period of overnight fasting with free access to water.

## Measurement of blood pressure and endothelium-dependent and -independent hypotensive responses

The rats were anesthetized using isoflurane (Forane, 2–4 percent in oxygen), to minimize depression of cardio-respiratory systems, and the right carotid artery and left jugular vein were cannulated to record the mean arterial pressure (MAP) and collect blood samples respectively. The MAP was recorded for 20 min using the Powerlab (AD Instruments Inc.) and Chart software, followed by recording of acetylcholine (ACh, Cat # A6625, Sigma-Aldrich, Oakville, ON, Canada) -induced endothelium dependent and sodium nitroprusside (SNP, Cat # S0501, Sigma-Aldrich, Oakville, ON, Canada) -induced endothelium-independent dose-related hypotensive responses. We administered bolus i.v. doses (0.125, 0.25, 0.5, 1, 2 and 4 µg/kg body wt.) of ACh and SNP *via* the jugular vein. The hypotensive responses to these agonists were recorded *via* the carotid artery cannula. The fall in mean MAP produced by these agonists was expressed as a percent of the basal MAP.

The blood samples were centrifuged at 12,000 rpm for 10 min to separate plasma, which was frozen and stored at -80 ˚C. The rats were then euthanized by cutting open the heart for exsanguination. Finally, organs and tissues were removed, rinsed with phosphate-buffered saline, frozen in liquid nitrogen, and stored at -80˚C. For the intestines we removed a 20 cm long piece of small intestine starting at about 5 cm from the gastroduodenal junction. It has been referred to as upper small intestine.

## Western blotting

Western blot was performed on tissue lysates to determine the enzyme expression as described previously [2]. The sample protein concentrations were determined with BioRad protein assay (Bio-Rad Laboratories Ltd., Mississauga, ON, Canada) and between 30 to 75 µg of protein were

loaded for different enzymes. The proteins were separated on 4–20 percent SDS-polyacrylamide pre-cast gels (Cat. # 456–1094, Bio-Rad Laboratories Ltd.) along with 5 μL of Precision Plus Dual Color Standards (Cat. # 161–0374, Bio-Rad Laboratories Ltd.), in the first well. The bands were transferred to a 0.45 μm polyvinylidene difluoride (PVDF) membrane (Cat. # 45004110, GE Healthcare Life Sciences, Mississauga, ON, Canada), and blocked with 5 percent bovine serum albumin (BSA, Cat. # A7906, Sigma-Aldrich Canada) for an hour. This was followed by primary antibody incubation at 4°C overnight (12 to 16 h) as follows: eNOS (1:500, Cat. # 611852, BD Transduction Laboratories, Mississauga, ON, Canada), SLC7A1 (CAT-1, 1:1000, Cat. # ABIN5965961, Antibodies-Online Inc., Atlanta, GA, USA), arginase I and arginase II (1:1000, Cat. # ab91279 and Cat. # ab203071, respectively), ADC (1:1000, Cat. # ab157214), agmatinase (1:1000, Cat. # ab231894), GATM (1:1000, Cat. # ab87062), all purchased from Abcam Inc., Toronto, ON, Canada. Horseradish peroxidase-conjugated secondary antibodies (Antimouse, 1: 10000, Cat. # 1706576, and Anti-rabbit, 1:10000, Cat. # 1706515, Bio-Rad Laboratories Ltd., Mississauga, ON, Canada) were incubated for 1 hour followed by Clarity Western Enhanced Chemiluminescence Blotting Substrate (Cat. # 1705061, Bio-Rad Laboratories Ltd.) to detect the immunoreactive proteins on the ChemiDoc Imaging System (G:BOX Chemi XX6, Syngene, Frederick, MD, USA). The protein bands on the image were manually quantified using GeneTools software (Syngene, Frederick, MD, USA). For loading control, the Invitrogen No-Stain Protein Labelling Reagent (Cat # A44449, Fisher Scientific) was used to perform total protein normalization.

## L-arginine assay

L-Arg levels were measured using a fluorometric Arginine Assay Kit (Cat. # ab252892, Abcam Inc., Toronto, ON, Canada), following kit instructions. The enzymatic assay in this kit only measures L-Arg. The fluorescence was measured with a fluorescence spectrophotometer (Fluoroskan Ascent, Thermo-Fisher Scientific, Vantaa, Finland) at 535/587 nm excitation/ emission wavelengths. The results obtained with this kit correlate with plasma L-Arg values in Sprague-Dawley rats using an amino acid analyzer [22], as well as HPLC [23, 24]. We could not find an assay that specifically measures D-Arg.

## Arginase activity assay

Arginase activity levels were measured with a colorimetric Arginase Activity Assay Kit (Cat. # ab180877, Abcam Inc., Toronto, ON, Canada), according to kit instructions. The optical density of the product immediately after adding the reaction mix was read at 570 nm in a kinetic mode for 30 min using a microplate spectrophotometer (Multiskan Spectrum, Thermo-Fisher Scientific, Vantaa, Finland).

## Urea and hydroxyproline assays

Urea levels were measured with a colorimetric Urea Assay Kit (Cat. # ab83362, Abcam Inc., Toronto, ON, Canada), according to kit instructions. The optical density was read at 570 nm in a spectrophotometer. Hydroxyproline levels were measured with a colorimetric Hydroxyproline Assay Kit (Cat. # ab222941, Abcam Inc., Toronto, ON, Canada), according to provided instructions. The absorbance was measured with a spectrophotometer at 560 nm.

## Nitric oxide synthase activity assay

NOS activity was measured with a colorimetric Nitric Oxide Synthase Activity Assay Kit (ab211083, Abcam Inc., Toronto, ON, Canada), according to included instructions. The absorbance was measured at 540 nm using a plate reader.

### Nitrate/Nitrite and L-citrulline assays

Nitrate plus nitrite levels were measured with a Nitrate/Nitrite Colorimetric Assay Kit (Cat. # 780001, Cayman Chemical, Ann Arbor, MI, USA) as per instructions provided. Nitrite levels were measured after conversion of nitrate to nitrite with nitrate reductase. One possibility to always remember is that microbial metabolism of unabsorbed L-Arg or D-Arg can produce nitrate/nitrite which can be absorbed and give erroneous values. The absorbance was measured at 540 nm on a microplate spectrophotometer. L-Citrulline levels were measured using a colorimetric assay kit following the provided protocol (Abcam, Mississauga, ON, Canada; Cat # ab242292).

### Creatinine and total polyamine assays

Creatinine levels were measured with a colorimetric Creatinine Assay Kit (Cat. # ab65340, Abcam Inc., Toronto, ON, Canada). The optical density was measured at 570 nm on a spectrophotometer. Total polyamine levels were measured with a fluorometric Total Polyamine Assay Kit (Cat. # ab239728, Abcam Inc., Toronto, ON, Canada). The fluorescence of the final product was measured with a fluorescence spectrophotometer (Fluoroskan Ascent, Thermo-Fisher Scientific) at 535/587 nm excitation/emission wavelengths.

### L-Lysine and asymmetric dimethylarginine assays

L-Lysine levels were measured with a fluorometric Lysine Assay Kit (Cat. # ab273311, Abcam Inc., Toronto, ON, Canada). The fluorescence was measured with a spectrophotometer at 535/587 nm excitation/emission wavelengths.

ADMA levels were measured with a colorimetric Asymmetric dimethylarginine ELISA Kit (Cat. #OKEH02587, Aviva Systems Biology, San Diego, CA, USA). Absorbance was measured at 450 nm using a microplate spectrophotometer.

### Statistical analysis

Our rats were ordered at the same time from the same batch and were randomly divided into three groups. We used the G*Power (v 3.1.9.7) software to perform an *A Priori* as well as post-hoc analysis to calculate power [complement of type 2 error probability (1-β)] to determine a minimal sample size and a power of >0.8. We used four most commonly used parameters of L-arginine supplement studies, *viz*. MAP, plasma nitrate+nitrite, plasma urea and plasma arginine. The mean values and SD for plasma nitrate, arginine and urea were from our previous study [2] and MAP values for *A Priori* calculation from our studies and studies reported in the literature. Nitrate values were significant whereas arginine and urea were not significant. MAP values for *A Priori* were significant as well as non-significant. The probability was $P = 0.05$. The actual power values were above 0.95 with total sample sizes ranging from 6 to 15, which justifies our total sample size of 18 (4+7+7). For analyzing the western blots and assay results, Graphpad PRISM software version 8 was used to perform one-way ANOVA with Tukey's as well Brown-Forsythe and Bartlett's post hoc tests, comparing between the rat treatment groups. The analysis results were expressed as Mean ± SEM.

## Results

### Oral L-Arg and D-Arg did not significantly affect the average body weight, mean arterial pressure, heart rate or the average daily water consumption

The 9-week-old rats were randomly allocated to one of three treatment groups and the average body weights of the three groups were not significantly different from each other at 9 wks. L-Arg

and D-Arg did not affect the age-related gain in average body weight at the end of 16 wks (Table 1). L-Arg and D-Arg were administered orally in drinking water, rather than parenterally, to mimic real life oral consumption of L-Arg supplements by people. Oral gavage for 16 wks was deemed to be very stressful and would increase the risk of mortality with pulmonary aspiration for the rats. The average daily water intake per rat was not significantly different among the three treatment groups at the beginning of treatment. L-Arg and D-Arg did not affect the average daily water intake at the end of the treatment period, compared to the control group (Table 1). L-Arg and D-Arg did not affect the mean arterial pressure or the heart rate (Table 1). 4 rats were allocated to the control group to reduce the number of animals in line with the three "Rs" of animal care. In hindsight, looking at the standard deviations in statistical analyses of some results, including the heart rate, it would have been prudent to include 7 rats in the control group. The daily L-Arg and D-Arg intake was maintained constant for each rat (Table 1).

## Oral L-Arg and D-Arg did not affect the acetylcholine- and sodium-nitroprusside-induced hypotensive responses in Sprague-Dawley rats

Treatment with oral L-Arg or D-Arg did not affect concentration-dependent hypotensive vasodilator responses elicited by i.v. bolus doses of the endothelium-dependent agonist acetylcholine or endothelium-independent agonist sodium nitroprusside in anaesthetized rats (Fig 1). The responses were analyzed at individual doses as well as the total area under curve for each treatment group. We have to admit here that *in vivo* hypotensive responses are not an ideal method to assess endothelial NO-mediated vasodilation and endothelial function, compared to isolated aortic rings or *in vivo* flow probes. *In vivo* hypotensive responses and MAP are influenced by many factors such as the baroreflex and autacoids and may not truly reflect pure NO-mediated vasodilation. However, we wanted to collect and freeze the aorta for various biochemical assays and could not risk leaving it at 37° C in an organ bath for 2–3 hours before freezing it.

## Oral L-Arg or D-Arg affected L-Arg levels in the skeletal muscle, upper small intestine and kidney, but did not affect L-Arg, L-lysine or asymmetric dimethylarginine levels in the plasma and different organs

As shown in Table 2, oral L-Arg treatment for 16 wks significantly increased L-Arg levels in the skeletal muscle, whereas oral D-Arg increased it in the upper small intestine and kidney,

**Table 1. Oral L-Arg and D-Arg did not affect the average body weight, mean arterial pressure, the heart rate or the average daily water intake of Sprague-Dawley rats.**

|  | Con | L-Arg | D-Arg |
|---|---|---|---|
| Average body wt. (g) at 9 wks | 368 ± 4 | 353±4 | 361 ± 5 |
| Average body wt. (g) at 25 wks | 759 ± 23 | 736 ±13 | 739 ± 14 |
| Average water intake (mL/d/rat) at 9 wks | 54 ± 2 | 51 ± 3 | 52 ± 4 |
| Average water intake (mL/d/rat) at 25 wks | 56 ± 8 | 66 ± 5 | 48 ± 2 |
| Daily arginine intake in drinking water (Please see the methods for calculation) | 0 mg/kg/d | 1000 mg/kg/d | 1000 mg/kg/d |
| Mean arterial pressure (mmHg) | 67 ± 6 | 78 ± 2 | 76 ± 2 |
| Heart rate (beats per min) | 232 ± 87 | 301 ± 11 | 336 ± 17 |

Nine-week-old male SD rats were treated with L-arginine (L-Arg) or D-arginine (D-Arg) at a dose of 1000 mg/kg/d each in drinking water for 16 wks. The control (Con) group received plain drinking water. For water intake, the rats were housed individually in separate cages and water intake was recorded every other day. The mean arterial pressure (MAP) was measured with an intra-arterial catheter and measured with Chart software on Powerlab, along with heart rate. The values are Mean ± SEM. ($n$ = 4 for Con and $n$ = 7 each for L-Arg and D-Arg groups).

## Endothelium-dependent and -independent hypotensive responses

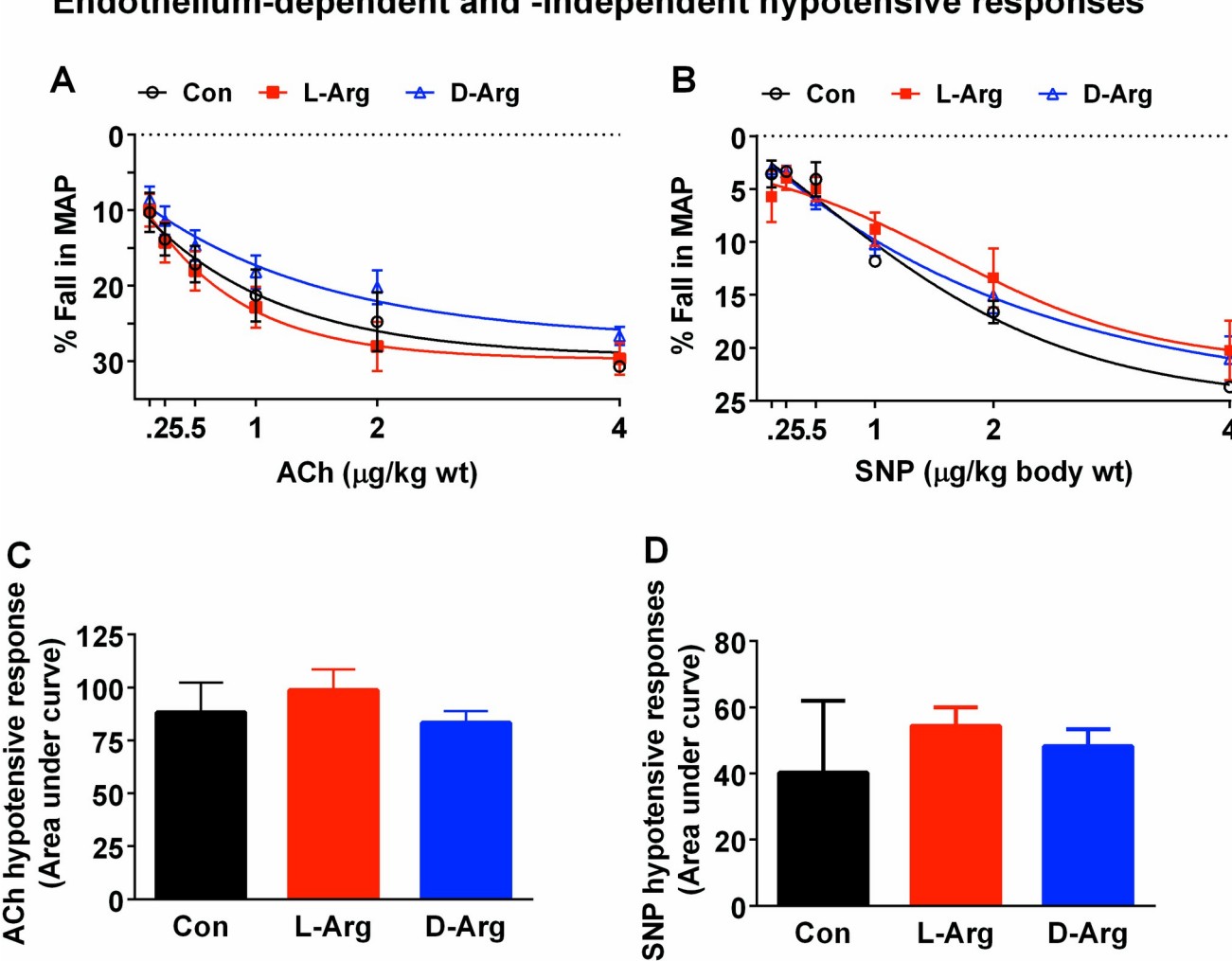

**Fig 1. Oral L-Arg and D-Arg did not affect the acetylcholine- and sodium-nitroprusside-induced hypotensive responses in Sprague-Dawley rats.** Nine-week-old male SD rats were treated with L-arginine (L-Arg) or D-arginine (D-Arg) at a dose of 1000 mg/kg/d each in drinking water for 16 wks. The control (Con) group received plain drinking water. The acetylcholine (ACh)-induced endothelium-dependent and sodium nitroprusside (SNP)-induced endothelium-independent concentration-related hypotensive responses were recorded as a fall in the mean arterial pressure (MAP) in anesthetized rats with a carotid artery cannula. The area under curve for each treatment group was calculated using GraphPad PRISM. The values are Mean ± SEM. ($n$ = 4 for Con and $n$ = 7 each for L-Arg and D-Arg groups).

compared to the control. L-Arg levels in the plasma, liver, aorta, lungs and brain of SD rats were not affected. The amount of collected urine varied significantly among the rats, ranging from 0.3 mL to 55 mL, which made the urinary measurements of various metabolites unreliable. Thereby, urinary L-Arg levels were not measured. Our oral dose for L-Arg and D-Arg was 1x times the arginine content in diet. While we did not measure the daily diet intake for each rat, we acknowledge that dietary L-Arg may have varied somewhat from day to day in each rat. The same will be true for humans taking fixed dose L-Arg supplements, in that their dietary arginine is likely to vary from day to day. L-lysine uses the same transporter as L-Arg, CAT-1, leading to competition in uptake for transport. Neither of the oral L-Arg or D-Arg treatment for 16 wks affected L-lysine levels in the plasma, liver, upper small intestine or

**Table 2. Oral L-Arg or D-Arg affected L-Arg levels in the skeletal muscle, upper small intestine and kidney, but did not affect L-Arg, L-lysine or asymmetric dimethylarginine levels in the plasma and different organs of Sprague-Dawley rats.**

| L-Arginine | Con | L-Arg | D-Arg |
|---|---|---|---|
| Plasma (μM) | 220 ± 22 | 194 ± 10 | 220 ± 9 |
| Liver (nmol/mg protein) | 8.1 ± 1.3 | 9.8 ± 0.9 | 9.5 ± 0.9 |
| Upper small intestine (nmol/mg protein) | 8.7 ± 0.4 | 7.6 ± 1.0 | 6.1 ± 0.3* |
| Kidney (nmol/mg protein) | 9.6 ± 0.6 | 13.3 ± 2.0 | 14.2 ± 0.6* |
| Aorta (nmol/mg protein) | 7.2 ± 1.0 | 4.9 ± 1.2 | 4.5 ± 0.6 |
| Lungs (nmol/mg protein) | 8.1 ± 1.8 | 7.4 ± 0.9 | 7.8 ± 0.9 |
| Brain (nmol/mg protein) | 4.2 ± 1.0 | 6.0 ± 3.0 | 4.8 ± 0.5 |
| Ske. muscle (nmol/mg protein) | 3.2 ± 0.8 | 6.5 ± 0.9 * | 4.9 ± 1.1 |
| **L-Lysine** | **Con** | **L-Arg** | **D-Arg** |
| Plasma (μM) | 86 ± 33 | 149 ± 107 | 73 ± 51 |
| Liver (nmol/mg protein) | 1.4 ± 0.2 | 1.5 ± 0.4 | 1.0 ± 0.1 |
| Upper small intestine (nmol/mg protein) | 7.7 ± 2.5 | 3.3 ± 0.3 | 4.7 ± 1.4 |
| Kidney (nmol/mg protein) | 1.7 ± 0.6 | 1.7 ± 0.7 | 1.5 ± 0.3 |
| **ADMA** | **Con** | **L-Arg** | **D-Arg** |
| Plasma (ng/mL) | 383 ± 84 | 283 ± 9 | 356 ± 16 |
| Liver (ng/mg protein) | 45 ± 9 | 34 ± 5 | 40 ± 4 |
| Upper small intestine (ng/mg protein) | 94 ± 14 | 80 ± 10 | 64 ± 5 |
| Lungs (ng/mg protein) | 153 ± 21 | 204 ± 28 | 190 ± 18 |
| Brain (ng/mg protein) | 323 ± 55 | 248 ± 30 | 281 ± 32 |

Nine-week-old male SD rats were treated with L-arginine (L-Arg) or D-arginine (D-Arg) at a dose of 1000 mg/kg/d each in drinking water for 16 wks. The control (Con) group received plain drinking water. L-Arg, L-lysine and asymmetric dimethylarginine (ADMA) levels were measured with specific assay kits. The values are Mean ± SEM. ($n$ = 4 for Con and $n$ = 7 each for L-Arg and D-Arg groups). (Ske. muscle–skeletal muscle).

*$P<0.05$ *vs.* respective control.

kidney (Table 2). Similarly, oral L-Arg or D-Arg did not significantly affect ADMA levels in the plasma, liver, upper small intestine, lungs and brain in comparison to the control group (Table 2). As mentioned before, in view of the large standard deviation of plasma ADMA levels in the control group it would have been prudent to include 7 animals in the control group as well.

## Oral D-Arg significantly increased cationic amino acid transporter 1 protein expression in the upper small intestine, but not in the liver or aorta

The expression of CAT-1 protein in the upper small intestine was significantly elevated in the D-Arg treatment group in comparison to the control group (Fig 2A, 2B). The expression of CAT-1 protein in the liver and the aorta was not significantly affected by either L- or D-Arg supplementation (Fig 2A and 2B).

## Oral L-Arg and/or D-Arg significantly affected arginase protein expression in the liver and the upper small intestine

Oral L-Arg and D-Arg increased arginase I protein expression in the liver, compared with the control group (Fig 3A, 3B). Arginase II protein expression in the upper small intestine was significantly reduced by L-Arg, compared to the control group (Fig 3A and 3B). Neither L-Arg or D-Arg affected arginase II protein expression in the kidney (Fig 3A and 3B).

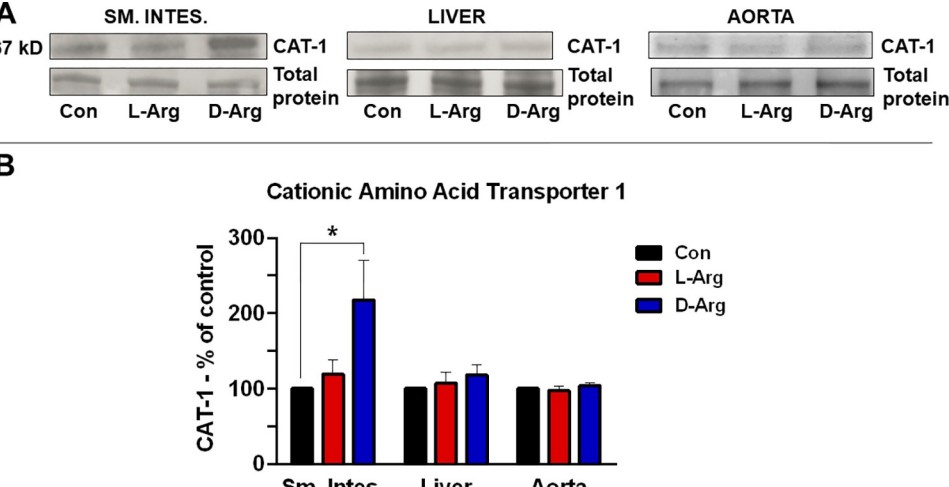

**Fig 2. Oral D-Arg increased the expression of cationic amino acid transporter 1 protein in the upper small intestine (Sm. Intes.).** Nine-week-old male Sprague-Dawley rats were treated with L-arginine (L-Arg) or D-arginine (D-Arg) at a dose of 1000 mg/kg/d each in drinking water for 16 wks. The control (Con) group received plain drinking water. Western blotting was performed using a specific anti-cationic amino acid transporter-1 [CAT-1 (SLC7A1), 1:1000] antibody as described in methods. The values are Mean ± SEM. ($n$ = 4 for Con and $n$ = 7 each for L-Arg and D-Arg groups). *$P$<0.05 *vs.* respective control.

## Oral L-Arg significantly decreased arginase activity in the plasma, but not in the upper small intestine, liver or kidney

Arginase activity in the plasma was significantly decreased by oral L-Arg, when compared with the control group (Fig 3C). However, L-Arg or D-Arg did not affect arginase activity in the upper small intestine, liver or kidney (Fig 3C).

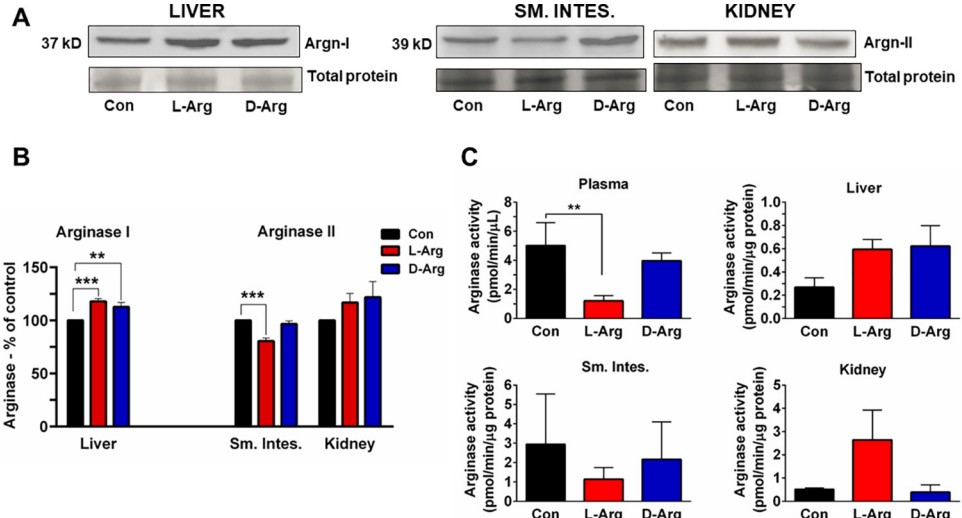

**Fig 3. Oral L-Arg and D-Arg increased arginase protein expression in the liver, but L-Arg decreased it in the upper small intestine (Sm. Intes.), whereas L-Arg decreased arginase activity in the plasma.** Nine-week-old male Sprague-Dawley rats were treated with L-arginine (L-Arg) or D-arginine (D-Arg) at a dose of 1000 mg/kg/d each in drinking water for 16 wks. The control (Con) group received plain drinking water. (A,B) Western blotting was performed using specific anti-arginase I and anti-arginase-II antibodies (1:1000). (C) An arginase activity assay was performed with a specific assay kit as described in methods. The values are Mean ± SEM. ($n$ = 4 for Con and $n$ = 7 each for L-Arg and D-Arg groups). **$P$<0.01, ***$P$<0.001 vs. respective control.

### Oral D-Arg significantly increased urea levels in the skeletal muscle

Oral D-Arg significantly increased urea levels in the skeletal muscle, compared to the control (Table 3), but L-Arg or D-Arg did not affect urea levels in the plasma, liver, upper small intestine and kidney (Table 3).

### Oral D-Arg significantly affected hydroxyproline levels in the upper small intestine and the brain

Oral D-Arg significantly decreased hydroxyproline levels in the upper small intestine compared to the control group (Table 3). However, in the kidney and the brain D-Arg treatment significantly increased hydroxyproline levels compared to the control group (Table 3). Neither of L-Arg or D-Arg treatment affected hydroxyproline levels in the plasma, liver, aorta or skeletal muscle in comparison to the control group (Table 3).

### Oral D-Arg significantly increased endothelial nitric oxide synthase protein expression in the aorta and kidney, but not in the brain

In both the aorta and kidney, eNOS protein expression was significantly elevated in comparison to the control group with oral D-Arg treatment for 16 wks, but not L-Arg (Fig 4). However, in the brain, neither L-Arg nor D-Arg significantly affected the expression of eNOS protein (Fig 4).

### Oral L-Arg or D-Arg did not affect nitric oxide synthase activity or L-citrulline levels plasma and /or different organs

As presented in Table 4, neither L-Arg nor D-Arg supplementation of SD rats for 16 wks altered NOS activity in the aorta, upper small intestine, liver, kidney, lungs, brain or skeletal muscle compared to the control group.

**Table 3. Oral D-Arg, but not L-Arg, significantly affected urea and hydroxyproline levels in some organs.**

| Urea | Con | L-Arg | D-Arg |
|---|---|---|---|
| Plasma (nM) | 69 ± 6 | 72 ± 4 | 67 ± 4 |
| Liver (nmol/mg protein) | 2.1 ± 0.5 | 2.1 ± 0.2 | 2.2 ± 0.2 |
| Upper small intestine (nmol/mg protein) | 2.0 ± 0.3 | 2.4 ± 0.4 | 0.9 ± 0.4 |
| Kidney (nmol/mg protein) | 2.7 ± 0.1 | 2.6 ± 0.1 | 2.6 ± 0.1 |
| Skeletal muscle (nmol/mg protein) | 2.4 ± 0.5 | 1.4 ± 0.5 | 4.3 ± 0.4* |
| **Hydroxyproline** | **Con** | **L-Arg** | **D-Arg** |
| Plasma (μg/mL) | 256 ± 18 | 246 ± 6 | 238 ± 13 |
| Liver (μg/mg protein) | 21 ± 3 | 21 ± 2 | 22 ± 2 |
| Upper small intestine (μg/mg protein) | 49 ± 5 | 52 ± 3 | 19 ± 6** |
| Kidney (μg/mg protein) | 5 ± 3 | 15 ± 6 | 26 ± 6* |
| Aorta (μg/mg protein) | 79 ± 5 | 87 ± 13 | 98 ± 11 |
| Brain (μg/mg protein) | 26 ± 4 | 28 ± 6 | 76 ± 7*** |
| Skeletal muscle (μg/mg protein) | 11 ± 2 | 22 ± 4 | 32 ± 9 |

Nine-week-old male Sprague-Dawley rats were treated with L-arginine (L-Arg) or D-arginine (D-Arg) at a dose of 1000 mg/kg/d each in drinking water for 16 wks. The control (Con) group received plain drinking water. Urea and hydroxyproline levels were measured with specific assay kits as described in methods. The values are Mean ± SEM. $n = 4$ for Con and $n = 7$ each for L- and D-Arg.

*$P<0.05$

**$P<0.01$ &

***$P<0.001$ *vs* respective control.

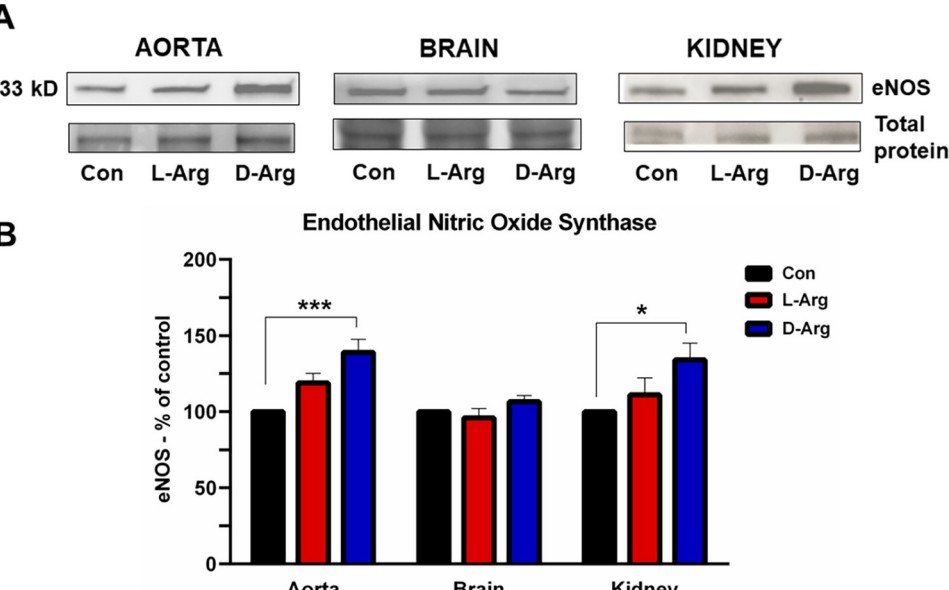

**Fig 4. Oral D-Arg increased endothelial nitric oxide synthase protein expression in the aorta and the kidney.**
Nine-week-old male Sprague-Dawley rats were treated with L-arginine (L-Arg) or D-arginine (D-Arg) at a dose of
1000 mg/kg/day each in drinking water for 16 wks. The control (Con) group received plain drinking water. Western
blotting was performed using specific anti-endothelial nitric oxide synthase (eNOS) antibody (1:500). The values are
Mean ± SEM. ($n$ = 4 for Con and $n$ = 7 each for L-Arg and D-Arg groups). $P<0.05$, ***$P<0.001$ vs. respective control.

Similarly, L-Arg nor D-Arg supplementation for 16 wks did not affect L-citrulline, a prod-
uct of NOS catalyzed L-Arg, levels in the plasma, liver, upper small intestine, kidney or skeletal
muscle compared to the control group (Table 4).

**Table 4. Oral L-Arg or D-Arg did not affect the nitric oxide synthase activity or L-citrulline levels in plasma and/or different organs of Sprague-Dawley rats.**

| Nitric oxide synthase activity (pmol/min/µg protein) | Con | L-Arg | D-Arg |
|---|---|---|---|
| Aorta | 0.057 ± 0.006 | 0.106 ± 0.027 | 0.058 ± 0.004 |
| Liver | 0.043 ± 0.008 | 0.054 ± 0.006 | 0.048 ± 0.003 |
| Upper small intestine | 0.061 ± 0.007 | 0.055 ± 0.007 | 0.066 ± 0.017 |
| Kidney | 0.040 ± 0.003 | 0.048 ± 0.003 | 0.046 ± 0.004 |
| Lungs | 0.062 ± 0.09 | 0.061 ± 0.004 | 0.065 ± 0.008 |
| Brain | 0.078 ± 0.013 | 0.068 ± 0.010 | 0.075 ± 0.016 |
| Skeletal muscle | 0.057 ± 0.018 | 0.079 ± 0.008 | 0.060 ± 0.006 |
| **L-citrulline** | **Con** | **L-Arg** | **D-Arg** |
| Plasma (mM) | 1.9 ± 0.2 | 2.4 ± 0.1 | 2.0 ± 0.1 |
| Liver (nmol/mg protein) | 27.6 ± 7.7 | 17.6 ± 1.5 | 21.0 ± 2.8 |
| Sm. Intes. (nmol/mg protein) | 115.8 ± 14.9 | 159.5 ± 38.5 | 199.0 ± 44.3 |
| Kidney (nmol/mg protein) | 63.4 ± 15.3 | 88.2 ± 9.2 | 79.7 ± 14.3 |
| Ske. Muscle (nmol/mg protein) | 202.2 ± 112.2 | 214.6 ± 28.2 | 274.1 ± 74.8 |

Nine-week-old male Sprague-Dawley rats were treated with L-arginine (L-Arg) or D-arginine (D-Arg) at a dose of 1000 mg/kg/day each in drinking water for 16 wks.
The control (Con) group received plain drinking water. Nitric oxide synthase (NOS) activity assay and L-citrulline levels were measured with specific assay kits. The
values are Mean ± SEM. $n$ = 4 for Con and $n$ = 7 each for L- and D-Arg.

### Oral L-Arg and/or D-Arg significantly affected nitrate and nitrite levels in the urine, upper small intestine and brain

Nitrite levels were significantly decreased in the urine in both the L-Arg and D-Arg groups compared to the control group (Fig 5B). L-Arg significantly increased nitrite levels in the brain whereas D-Arg decreased it in the upper small intestine, compared to control (Fig 5H and 5C). Nitrite levels in the plasma, liver, kidney, lungs, and skeletal muscle were not affected by either L-Arg or D-Arg (Fig 5).

### Oral L-Arg and/or D-Arg significantly affected the arginine:glycine amidinotransferase protein expression in the liver and the kidney

AGAT protein expression was significantly reduced in the liver following treatment with both L-Arg and D-Arg, compared to the control group (Fig 6A and 6B). However, in the kidney, L-Arg significantly increased AGAT protein expression compared to the control group (Fig 6A and 6B). In the upper small intestine and brain, neither L-Arg nor D-Arg affected AGAT protein expression (Fig 6).

### Oral L-Arg and D-Arg significantly affected creatinine levels in the urine and the upper small intestine

Creatinine levels were significantly reduced in the urine of L-Arg and D-Arg treated groups of SD rats, compared to the control group (Table 5). D-Arg treatment also significantly increased creatinine levels in the upper small intestine compared to the control group (Table 5). However, creatinine levels in the plasma, liver and kidney were not affected by L-Arg or D-Arg treatment (Table 5).

### Oral L-Arg and D-Arg significantly affected the arginine decarboxylase protein expression in the liver and upper small intestine

ADC protein expression was significantly increased in the liver following 16-week treatment with oral L-Arg, as well as D-Arg in comparison to the control group (Fig 7A). Both L-Arg and D-Arg also significantly decreased the expression of ADC protein in the upper small intestine, compared to the control group (Fig 7A). However, in the kidney and brain, neither L-Arg or D-Arg had any significant effects on ADC protein expression (Fig 7A).

### Oral D-Arg significantly affected the agmatinase protein expression in the kidney, upper small intestine and brain

Treatment for 16 wks with oral D-Arg, but not L-Arg, significantly increased agmatinase protein expression in the kidney and the upper small intestine, compared to the control group (Fig 7B). However, in the brain, D-Arg significantly decreased agmatinase protein expression, compared to the control group (Fig 7B). In the liver, neither L-Arg or D-Arg affected the expression of agmatinase protein (Fig 7B).

### Oral L-Arg significantly increased total polyamines levels in the plasma

The levels of total polyamines including putrescine, spermine and spermidine, were significantly increased in the plasma for the L-Arg treated group of SD rats, compared to the control group (Table 6). Neither of L-Arg or D-Arg affected the total polyamine levels in the liver, upper small intestine, kidney or skeletal muscle of SD rats (Table 6).

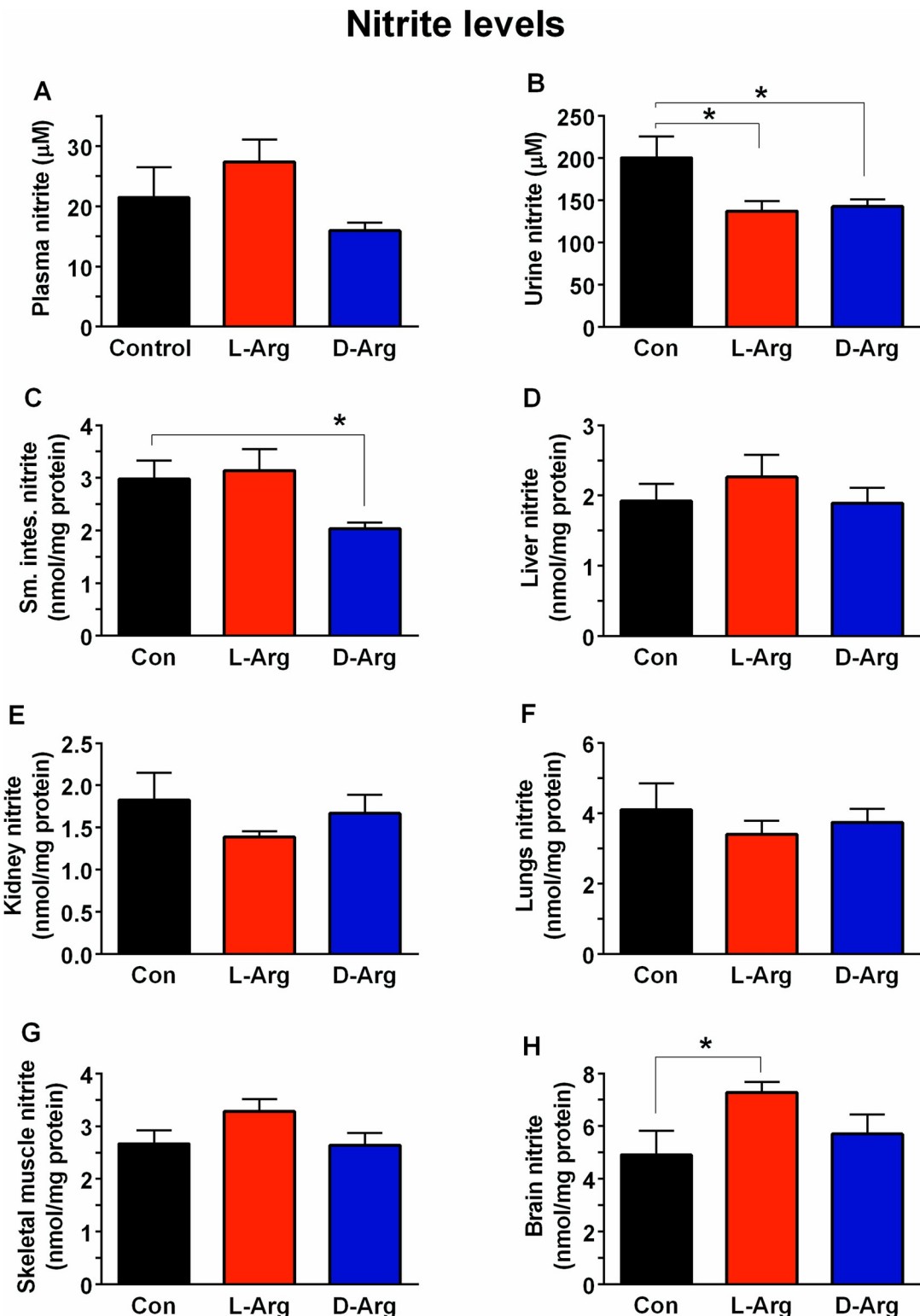

**Fig 5. Oral L-Arg and/or D-Arg significantly affected nitrite levels in the urine, upper small intestine and brain.** Nine-week-old male Sprague-Dawley rats were treated with L-arginine (L-Arg) or D-arginine (D-Arg) at a dose of 1000 mg/kg/d each in drinking water for 16 wks. The control (Con) group received plain drinking water. Nitrate plus nitrite levels were measured as nitrite after conversion of nitrate to nitrite with nitrate reductase and measured with a specific assay kit. The values are Mean ± SEM. ($n$ = 4 for Con and $n$ = 7 each for L-Arg and D-Arg groups). *$P$<0.05 vs. respective control.

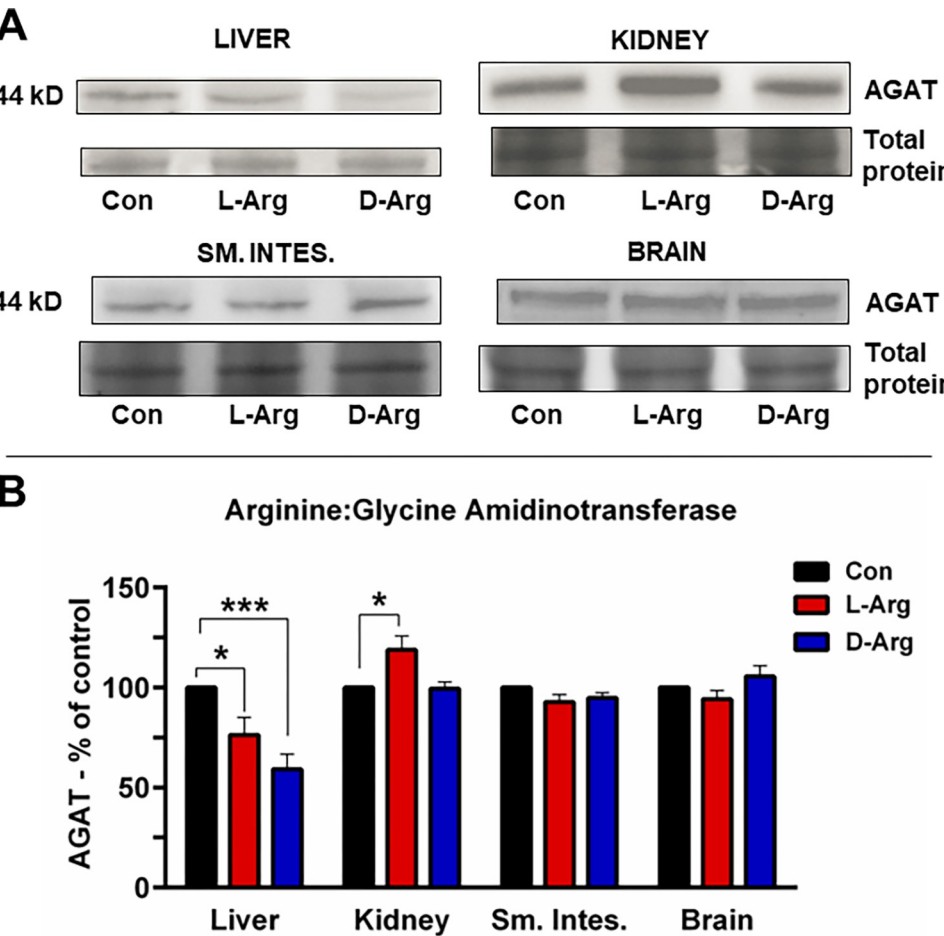

**Fig 6. Oral L-Arg and D-Arg decreased arginine:glycine amidinotransferase protein expression in the liver, but L-Arg increased it in the kidney.** Nine-week-old male Sprague-Dawley rats were treated with L-arginine (L-Arg) or D-arginine (D-Arg) at a dose of 1000 mg/kg/d each in drinking water for 16 wks. The control (Con) group received plain drinking water. Western blotting was performed using specific anti-arginine:glycine amidinotransferase (AGAT) antibody (1:1000). The values are Mean ± SEM. $n = 4$ for Con and $n = 7$ each for L-Arg and D-Arg. $^{*}P<0.05$, $^{***}P<0.001$ *vs*. respective control.

**Table 5. Oral L-Arg and D-Arg decreased creatinine levels in the urine, but D-Arg increased it in the upper small intestine.**

| Creatinine | Con | L-Arg | D-Arg |
|---|---|---|---|
| Plasma (mM) | 2.0 ± 0.3 | 2.3 ± 0.3 | 1.9 ± 0.2 |
| Urine (mM) | 44 ± 13 | 18 ± 5* | 15 ± 3* |
| Liver (nmol/mg protein) | 62 ± 12 | 58 ± 8 | 40 ± 5 |
| Upper small intestine (nmol/mg protein) | 157 ± 3 | 144 ± 12 | 213 ± 12* |
| Kidney (nmol/mg protein) | 66 ± 13 | 55 ± 7 | 52 ± 7 |

Nine-week-old male Sprague-Dawley rats were treated with L-arginine (L-Arg) or D-arginine (D-Arg) at a dose of 1000 mg/kg/d each in drinking water for 16 wks. The control (Con) group received plain drinking water. Creatinine levels were measured with a specific assay kit. The values are Mean ± SEM. *n* in brackets for each group. $n = 4$ for Con and $n = 7$ each for L-Arg and D-Arg.

$^{*}P<0.05$ *vs* respective control.

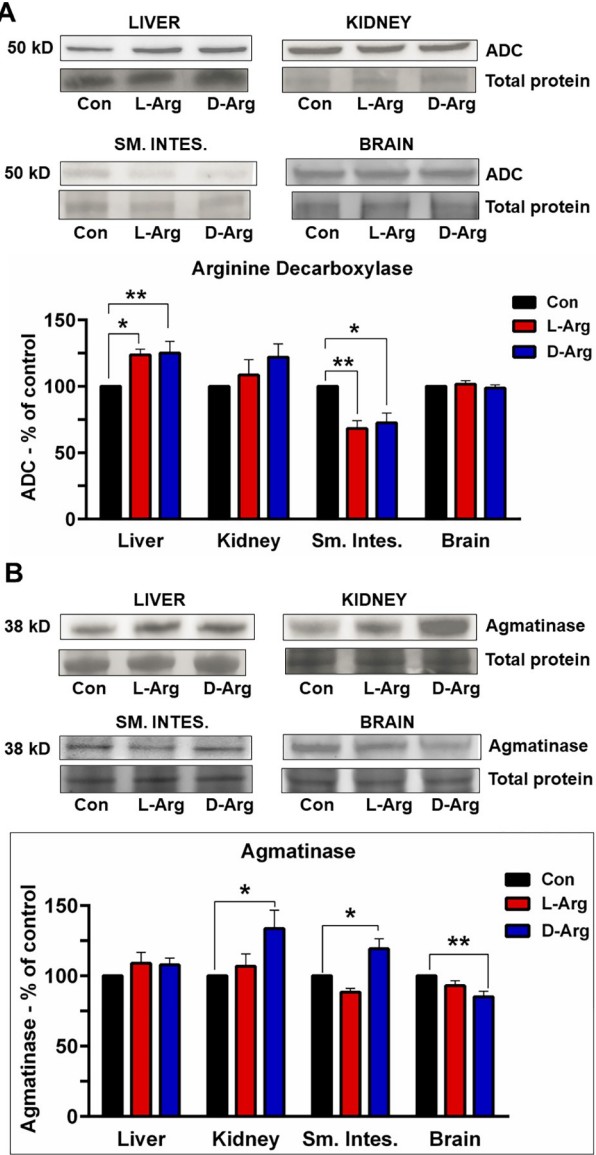

**Fig 7. Oral L-Arg and D-Arg increased arginine decarboxylase protein expression in the liver, but decreased it in the upper small intestine (Sm. Intes.); whereas D-Arg increased agmatinase protein expression in the kidney and the upper small intestine, but decreased it in the brain.** Nine-week-old male Sprague-Dawley rats were treated with L-arginine (L-Arg) or D-arginine (D-Arg) at a dose of 1000 mg/kg/d each in drinking water for 16 wks. The control (Con) group received plain drinking water. Western blotting was performed using specific anti-arginine decarboxylase (ADC) (A) and anti-agmatinase (B) antibodies (1:1000). The values are Mean ± SEM. $n$ = 4 for Con and $n$ = 7 each for L-Arg and D-Arg. *$P<0.05$, **$P<0.01$ *vs.* respective control.

## Discussion

We have recently reported the effects of oral L-Arg and D-Arg at 500 mg/kg/d for 4 wks in male SD rats [2]. Here we report the effects of a higher dose of oral L-Arg and D-Arg at 1000 mg/kg/d for a longer duration of 16 wks in 9 wk old male SD rats. Our results reveal complexities in L-Arg and D-Arg metabolism and seeming adaptations to oral supplementation at a different dose for a different duration. We expected the effects of 1000 mg/kg/d to be replicated at a magnified level compared with a dose of 500 mg/kg/d. Instead, we are reporting

**Table 6. Oral L-Arg increased total polyamine levels in the plasma.**

| Total polyamines | Con | L-Arg | D-Arg |
|---|---|---|---|
| Plasma (μM) | 23.5 ± 1.7 | 29.8 ± 1.6* | 22.8 ± 1.0 |
| Liver (nmol/mg protein) | 16.3 ± 3.4 | 17.4 ± 2.3 | 14.0 ± 1.1 |
| Upper small intestine (nmol/mg protein) | 7.3 ± 1.1 | 8.5 ± 1.0 | 6.9 ± 0.4 |
| Kidney (nmol/mg protein) | 3.8 ± 0.3 | 3.2 ± 0.3 | 4.0 ± 0.3 |
| Ske muscle (nmol/mg protein) | 12.2 ± 1.2 | 17.6 ± 2.1 | 13.8 ± 0.9 |

Nine-week-old male Sprague-Dawley rats were treated with L-arginine (L-Arg) or D-arginine (D-Arg) at a dose of 1000 mg/kg/d each in drinking water for 16 wks. The control (Con) group received plain drinking water. Total polyamines levels were measured with a specific assay. The values are Mean ± SEM. $n$ in brackets for each group. $n$ = 4 for Con and $n$ = 7 each for L-Arg and D-Arg. *$P<0.05$ *vs* respective control.

differences between the two doses and durations which underscore the need for further testing with a couple more different doses and durations to fully understand the impact of oral arginine supplements and enable their safe and effective use.

In terms of human equivalent dose, a dose of 1000 mg/kg/d is equivalent to a dose of about 162 mg/kg/d for humans (approx. 11.3 g/d for a 70 kg adult), according to the conversion guide provided by the Food and Drug Administration [25]. The purpose of this guidance is to determine the maximum recommended starting dose for new molecular entities for first human clinical trials, based on doses of that entity used in animal studies, without causing toxicity. Based on the FDA algorithm provided, it is stated that for conversion the rat dose be multiplied by 0.16 or divided by 6.2 to convert it to a human equivalent dose. This rat dose is considered as a moderate human dose because a meta-analysis that included 11 trials studying the effect of L-Arg supplements on blood pressure in humans used L-Arg doses ranging from 4 to 24 g/day for periods of 2 to 24 wks [18]. Male rats were chosen to eliminate possible confounding effects of female sex hormones on L-Arg metabolism. Further studies using female rats will be carried out separately.

There was no overt toxicity or mortality in the rats after oral L-Arg or D-Arg at 1000 mg/kg/d for 16 wks. The activities and feeding behaviour of all the rats, which were observed every other day, were normal. As with the previous study [2], the body weight of the SD rats treated with L-Arg or D-Arg was not significantly different from control group, and the age-related increase in body wt was not affected (Table 1). These consistent findings can potentially suggest that oral arginine does not affect the appetite or food consumption. Since arginine was administered in drinking water, we measured the daily water intake for each rat and adjusted the dose accordingly every other day. The daily average water intake was similar among the three groups of 9-week-old SD rats at the start of the treatment (Table 1). However, unlike the 500 mg/kg/d dose [2], at the end of the 16-week treatment period, the D-Arg group showed significantly less water intake on average per day when compared to the L-Arg group, but not the control group (Table 1). One possibility could be a mild distinct odour and a light yellow colour of the D-Arg solution, compared with the colourless and odourless L-Arg solution, despite the same 7.4 pH for both. The chemical reason for this difference could not be determined.

L-Arg and D-Arg did not affect the MAP or the heart rate in SD rats (Table 1), thereby reinforcing the results on MAP obtained in our previous study with a lower dose and shorter treatment duration [2]. MAP is regulated by multiple factors in the body such as the baroreflex mediated by the sympathetic and parasympathetic systems, the renin-angiotensin aldosterone system (RAAS), oxygen levels [26] and locally produced mediators such as NO [27] and endothelin [28]. Joyner and Limberg [26] have proposed that blood pressure is regulated

around a fixed set point which is not absolute, but this set point changes under different physiological and pathological situations and at different ages. In healthy rats or people these regulatory mechanisms might be negating any long-term changes in MAP produced by L-Arg and its vasodilatory effect. This questions the presumed effectiveness of oral L-Arg supplements *in vivo* as it relates to MAP, but not its acute vasodilatory effect observed as a hypotensive response, especially in healthy people. The same reasoning goes for no changes in the heart rate by L-Arg and D-Arg (Table 1). However, oral L-Arg significantly reduced the MAP in Zucker Diabetic Fatty rats, at a dose of 1000 mg/kg/d for 12 wks, in a study performed in our lab (unpublished results). Thus, L-Arg supplements might be beneficial in disease conditions such as hypertension [18], hypercholesterolemia [19] and diabetes [20, 29], and not in healthy normotensive people, a view supported by a study by Ast *et al.* [30] in their study on 19 healthy subjects who received 6 or 12 g/d of L-Arg for 4 wks. In disease conditions one or more of the regulatory mechanisms are altered, either at the cellular level or at the whole body level, such as increased sympathetic and increased RAAS activity and increased oxidative stress and decreased NO availability in essential hypertension [31], diabetes [32, 33], atherosclerosis [34], aging [35] etc., which may be partially corrected by L-Arg supplements, especially if there is increased oxidative stress and reduced NO availability, with a decrease in MAP. The baroreflex and chemoreflex and their interactions are altered differently in $N^G$-nitro-L-arginine (L-NAME), an inhibitor of NOS, induced hypertension, wherein the hypertension is mainly sympathetically and resistance vessel mediated with complete inhibition of cardiac vagal activity, resulting in exaggerated vagal tone whenever the baroreflex is activated [36]. These authors [36] hypothesized that the baroreflex and chemoreflex and their interactions are altered in pathological conditions. There is altered pressure-volume regulation or abnormal pressure natriuresis in hypertension wherein the same amount of sodium is excreted despite elevated blood pressure, as in normotension [37].

Oral L-Arg and D-Arg also did not affect acetylcholine-induced endothelium-dependent and sodium nitroprusside-induced endothelium-independent hypotensive responses (Fig 1). It is worth pointing out that *in vivo* hypotensive responses produced by a vasodilator are momentary acute changes in recorded MAP, which are different than a recording of stable compensated MAP obtained at any given time. However, as explained in the methods, *in vivo* hypotensive responses will reflect vasodilation influenced by several factors including the acute baroreflex compensation and autacoids, and not just NO-mediated vasodilation, which is a limitation of these results. Oral L-Arg has been shown to significantly increase flow-mediated dilation of brachial artery in patients with essential hypertension [38]. One reason why L-Arg supplements may not benefit healthy people is because the intracellular concentration of L-Arg is several hundred μM which far exceeds the ~5 μM $K_M$ of NOS [39]. The 'arginine paradox', where *in vivo* L-Arg supplements lead to more NO production despite enough intracellular L-Arg concentrations, might be mediated by insulin-induced vasodilation, which is released by L-Arg [40]. Another reason may be because arginases and NOSs compete for L-Arg and the catalytic activity of arginases is more than 1000 times that of NOS [5], thereby diverting L-Arg to the arginase metabolic pathway and creating NO deficiency. On the other hand, the beneficial effects of L-Arg supplements in disease conditions can be explained in a situation where there are elevated levels of ADMA, such as in renal failure [41], preeclampsia [42], high cholesterol levels [43] and in older age (>70 yrs) [44]. We also measured ADMA levels in the plasma, liver, ileum, lungs and brain and found that they were not affected by oral L- or D-Arg (Table 2). ADMA is synthesized during protein turnover and posttranslational modification where L-Arg gets methylated. ADMA has been reported as an endogenous eNOS inhibitor, with its levels being elevated in renal failure [41]. ADMA can compete with L-Arg and create NO deficiency, which can be overcome with excess L-Arg from supplements. At the

same time L-Arg supplements should not increase ADMA levels. It is worth noting that in our previous study with a 500 mg/kg/d dose, L-Arg significantly increased the levels of ADMA in the kidney in comparison to both the control and D-Arg groups [2]. In one study on Watanabe hyperlipidemic rabbits, administration of L-Arg at 1.5% in 1 g/kg body weight/d chow diet for 8 wks resulted in a significant increase in the plasma L-Arg/ADMA ratio, with the authors concluding that the L-Arg/ADMA ratio positively correlates with the progression of athero-sclerotic plaques [45]. In another study on preeclamptic pregnant rats, administration of L-Arg orally at 21 mg/kg/d on 18[th] and 19[th] days of pregnancy resulted in an increase in plasma ADMA levels compared to control preeclamptic pregnant rats, but a significant decrease in blood pressure [46]. In another study on patients with mild hypertension, adminis-tration of L-Arg at 2 g/3 times a day or 4 g/3 times a day for 28 days caused a significant increase in plasma ADMA levels [47]. On the other hand a significant decrease in plasma ADMA levels, but an increase in L-Arg/ADMA ratio was reported by Lucotti et al. [48] in non-diabetic patients with cardiovascular disease who were treated with L-Arg at 6 g/d for 6 months. In one study oral L-Arg given to people over the age of 70, at a dose of 8 g twice a day for two wks, improved endothelial function and L-Arg/ADMA ratio, although ADMA levels remained unchanged [44]. Studies on the effects of arginine supplements on ADMA levels and L-Arg/ADMA ratio have produced mixed results, which may be due to differences in dose or duration of treatment, but are very necessary since ADMA levels are associated with disease conditions [49, 50]. It may be fair to conclude that a decrease in ADMA or an increase in L-Arg/ADMA ratio would be beneficial but the effect of L-Arg supplements on these two parameters is inconclusive and may be dose or treatment duration related.

Following oral administration, it is crucial to know its distribution in different organs and any changes in plasma and tissue levels. Oral L- and D-Arg did not significantly affect the nor-mal plasma levels or levels in most organs of L-Arg (Table 2), consistent with the findings of the lower dose of 500 mg/kg/d [2]. However, oral L-Arg increased levels in the skeletal muscle, for which we could not find any other literature report, whereas oral D-Arg increased them in the upper small intestine and kidney (Table 2). D-Arg might have prevented L-Arg metabo-lism in the upper small intestine to increase levels. There are several possible reasons for no change in plasma and tissue levels. Approximately 40% of oral arginine is metabolized in the gastrointestinal epithelium and liver before systemic absorption [51]. Even after absorption it is rapidly metabolized as suggested by a half-life of about 1 hour [52, 53]. This suggests fast and dynamic metabolism of arginine and the possibility that the body might be capable of rap-idly adjusting to dietary fluctuations or oral supplements, especially with multiple metabolic pathways. Also, the absorption from drinking water would be in small amounts throughout the day, which is constantly getting metabolized, with no large plasma level peaks. Administra-tion of L-Arg orally as a single bolus dose of 500 mg/kg to diabetic and non-diabetic BioBreed-ing rats resulted in a peak plasma concentration which was 2.2 fold and 3 fold higher, respectively, than at baseline1 h after administration [52]. The average absolute oral bioavail-ability was 0.64 and 0.60 in non-diabetic and diabetic BB rats, respectively [52]. This suggests significant first pass metabolism even after a large oral bolus dose. In the same study [52] administration of L-Arg in drinking water at 2.14 g/kg/d for 10 wks was reported to cause an increase in serum concentration, although values were not provided. It should be noted that they used an oral dose (2.14 g/kg/d) [52] twice as much as the current study. In a study in healthy human volunteers a single oral bolus dose of L-Arg at 10 g/100 mL water resulted in a three-fold increase in peak plasma L-Arg concentration 1 hour after administration, with a large interindividual variation [54]. In another study on 12 healthy human volunteers, the mean serum L-Arg level was not different than baseline after a daily oral dose of 3 g/d for 1 wk, whereas it was significantly higher than baseline with a dose of 9 g/d for 1 wk with no

further increase at 21 g/d for 1 wk [55]. These results suggest that very high oral doses for a week or more may increase plasma L-Arg concentration but lower doses may fail to do so, possibly because of significant first pass metabolism.

Another potential explanation for no significant changes in arginine levels might have been a downregulation of the arginine transporter CAT-1 in the ileum [56], a primary site for its absorption, which was observed in our previous study with a lower dose [2], but not in the current study with a higher dose (Fig 2). CAT-1 is a primary transporter for arginine [57]. Surprisingly, CAT-1 expression in the ileum was significantly elevated with D-Arg treatment and not L-Arg, when compared to control (Fig 2). One possible reason for the increased CAT-1 expression in the D-Arg group in ileum could be that supplemental D-Arg might be outcompeting the dietary L-Arg for transport causing an adaptive increase in CAT-1 expression. CAT-1 expression was also measured in the liver, another important site of arginine transport, where L- or D-Arg had no effect. CAT-1 expression in the aorta, where it is co-localized with eNOS [58], was also not affected by L- or D-Arg (Fig 2). Other studies have reported a downregulation of CAT-1 by oral arginine supplements and likely reflects an adaptation to regulate absorption in response to increased oral load [2, 59]. To the best of our knowledge, the effects of D-Arg on CAT-1 have not been reported. It might have been useful to measure the concentration of L-Arg and D-Arg in the mucosa of the proximal small intestine.

Since CAT-1 also transports L-lysine, we measured L-lysine levels in the plasma and selected organs, which were not significantly affected by either oral L-Arg or D-Arg (Table 2). This correlates with the unchanged CAT-1 expression in the ileum in the L-Arg group, but not with the increased expression of CAT-1 in the D-Arg group (Fig 2). In our previous study with 500 mg/kg/d for 4 wks, plasma L-lysine levels were significantly lower in the L-Arg group compared with the D-Arg group, which can be explained with decreased CAT-1 expression in the liver and ileum of L-Arg treated rats [2]. L-Arg has been shown to reduce the absorption of L-lysine depending on the dose of L-Arg administered in horses [60], and also from caecum when the two amino acids are administered together because the ability of arginine absorption is greater than that of lysine [61]. Thus, if L-Arg is significantly reducing the absorption of L-lysine, an essential amino acid, then it is a matter of concern and lysine levels should be checked when oral L-Arg supplements are taken.

Arginase, a principal enzyme for arginine metabolism, produces urea for ammonia detoxification [5]. In the upper small intestine arginase II protein expression was significantly reduced with oral L-Arg compared to the control (Fig 3), which partly aligns with our previous study where both L- and D-Arg at 500 mg/kg/d reduced arginase II protein expression [2]. Arginase I expression in the liver was significantly elevated by both oral L-Arg and D-Arg as compared with control (Fig 3), which is different from our previous study with 500 mg/kg/d, where D-Arg reduced the arginase I expression in the liver compared to control [2]. The increase in liver arginase may possibly be in part due to reduced arginase in the upper small intestine, which would then allow more arginine to go to the liver. This finding also aligns with the report that exogenous arginine induces arginase and offers one major reason why oral arginine supplements may not be beneficial [62]. This can be problematic because upregulation of arginase attenuates the formation of NO by competing with eNOS and contributing to conditions such as hypertension and endothelial dysfunction [9, 10]. Oral L- and D-Arg at 1000 mg/kg/d had no significant effect in the kidney (Fig 3), which aligns with our previous study with 500 mg/kg/d [2]. This could possibly be due to sufficient arginine not reaching the kidney after oral absorption. In a study with oral L-Arg (2.25 percent in drinking water for 6 wks), it did not affect arginase protein expression in the liver or the kidney, in young and aged rats [63]. Elevated plasma arginine levels in hyperargininemia have been shown to increase arginase I and II protein expression [64]. Arginase I and II are also differentially regulated [63,

64]. Increased expression or activity of arginase and polyamine levels have been reported in several cancer types [65] and this is of serious concern if L-Arg supplements consistently increase arginase expression/activity. As to why an excess of a substrate would upregulate one enzyme and down-regulate another one may be related to multiple levels and mechanisms of regulation, which are unique for each enzyme.

As for arginase activity, oral L-Arg decreased it in the plasma but not in the ileum, liver and kidney (Fig 3C). In comparison, L- and D-Arg did not affect arginase activity in the plasma and organs with a 500 mg/kg/d dose [2]. One possibility for decreased arginase activity in the plasma could be acute peaks in plasma arginine levels with absorption, which would decrease activity. In contrast, a study has reported that treatment of aged (22–24 wk old) Wistar rats with 2.25% L-Arg in drinking water for 6 wks normalized the decreased plasma arginine levels and decreased arginase activity. However, in this study arginase activity was measured as arginine/ornithine ratio, which is not reliable, and aging was a factor in the increased arginase activity [63].

To assess the impact of supplements on the arginase pathway, we measured the levels of its major metabolites, urea and hydroxyproline, in the plasma and various organs (Table 3).

Urea levels were not altered with oral L- or D-Arg in the plasma, liver and kidney in comparison to control and with each other (Table 3). However, D-Arg significantly reduced urea in the ileum compared with the L-Arg group, but increased it in the skeletal muscle compared with the L-Arg group (Table 3). This effect of D-Arg was not expected and is perplexing. The significance of the D-Arg effect can be questioned as it was not in comparison with control. Interestingly in our previous study, D-Arg at 500 mg/kg/d for 4 wks had significantly increased urea levels in the liver and the kidney, compared to the control group [2]. At the whole body level 60% of dietary L-Arg, regardless of the amount taken in, was converted into urea, and it occurred during the first pass [14]. These authors [14] propose a tight metabolic compartmentalization of arginine metabolic pathways, which ensures homeostasis of NO production in healthy humans in the face of wide variations in L-Arg availability, a view which needs experimental evidence for support. Unfortunately, no clear explanation has been found so far, with limited research about the pharmacodynamics of oral arginine and its effects on the skeletal muscle related to urea production. Urea levels have been shown to negatively regulate arginase enzymatic activity [66].

The other product of L-Arg catalyzed by arginase is L-ornithine, which is metabolized by ornithine aminotransferase (OAT) to produce L-proline. We recognize that hydroxyproline, produced by hydroxylation of L-proline, is not a direct product of OAT. However, since hydroxyproline is a major component and indicator of collagen stability, we measured hydroxyproline levels, to determine if arginine supplements have any effects on collagen stability. Oral D-Arg significantly decreased it in the upper small intestine (Table 3) and increased it in the kidney and the brain, compared with the control (Table 3). However, in the plasma, liver, kidney, and the aorta, the levels of hydroxyproline were not affected. In our previous study [2], hydroxyproline levels in the plasma and organs were not affected by 500 mg/kg/d of L- or D-Arg. The effect of D-Arg on hydroxyproline levels is difficult to interpret and associate with the results on arginase expression and activity, due to limited knowledge on the physiological impact of this arginine isoform. However, it is important to note that in the ileum, the levels of both urea and hydroxyproline were significantly reduced with D-Arg treatment (Table 3). This may suggest that D-Arg downregulates the arginase pathway in the ileum to cause significantly lower production of its metabolites, although the expression and activity of arginase II remained unchanged compared to control (Fig 3). Hemodynamic stretch of vascular smooth muscle cells has been shown to regulate arginine metabolism and synthesis of L-proline and collagen [67]. In one study oral L-Arg supplementation at 30 g/d for 2 wks to 30

healthy human volunteers significantly increased hydroxyproline, as an index of collagen synthesis, and protein content in a surgical wound and improved wound healing [68]. In a study on rats intraperitoneal arginine injections at 1 g/kg/d in three divided doses for 10 days significantly increased hydroxyproline levels in the wound, and increased wound breaking strength, in a surgical wound combined with hemorrhagic shock [69]. Due to its possible usefulness in wound healing it is important to confirm the dose and treatment duration of L-Arg supplements which enhance hydroxyproline levels.

Another major enzyme for L-Arg metabolism is NOS, which produces NO, a vasodilator. Oral D-Arg, but not L-Arg, at 1000 mg/kg/d for 16 wks significantly increased eNOS protein expression in the aorta and kidney compared to the control (Fig 4), which may be because it competes with L-Arg for eNOS, thereby creating relative NO deficiency and upregulating expression. D-Arg is not a substrate for eNOS [70]. In the previous 500 mg/kg/d study [2], eNOS expression in the aorta was increased with L-Arg treatment, and both oral L- and D-Arg increased eNOS expression in the kidney in comparison with control. L-Arg has been shown to increase eNOS expression in hypercholesterolemic rabbit aorta [71]. In high salt diet-induced hypertensive rats administration of oral L-Arg at 100 mg/kg/d for 12 wks prevented the reduced eNOS mRNA expression in the aorta and increased NO levels [72]. In rabbits implanted with a flow-diverting device in the aorta, treatment with L-Arg at 2.25% in drinking water for 8 wks increased eNOS protein expression in the aorta and reduced vascular contractility [73]. In streptozotocin-induced diabetic [74] mice, L-Arg administration at 1.5% in drinking water for 9 wks, did not prevent the significantly decreased kidney eNOS protein expression. Thus, overall it appears that L-Arg supplementation increases aortic eNOS expression. L-Arg availability and endothelial arginase I have been shown to regulate eNOS protein expression [75]. The reason why a higher dose and a longer duration of L-Arg treatment would not have any effect on eNOS protein expression in the aorta or kidney is a matter of conjecture, and could possibly be due to the body adjusting more with higher dose and longer treatment duration, which hopefully will be clarified with the third phase of the study with a dose of 500 mg/kg/d for 16 wks.

The activity of NOS was not affected by L-Arg or D-Arg in various organs such as the aorta, ileum, liver, kidney, lungs, brain and skeletal muscle (Table 4). This almost aligns with no significant effect of a lower dose of L- and D-Arg on NOS activity in our previous study, except in the lungs where it was significantly decreased with D-Arg compared to the control and L-Arg groups [2], and also reported in humans in other studies [76, 77]. Thus, increased eNOS protein expression, rather than enzyme activity, appears to commonly compensate for reduced NO production in different conditions. Nitrate and nitrite are the main stable metabolic products of NO. Nitrite levels were measured after conversion of nitrate to nitrite with nitrate reductase. Nitrite levels were significantly increased in the plasma of the L-Arg group in comparison with the D-Arg group, but not the control (Fig 5), which questions its relevance. In our previous study [2], L-Arg at 500 mg/kg/d for 4 wks significantly elevated plasma nitrite levels compared to both control and D-Arg groups. L-Arg incubated with cultured human endothelial cells also increased nitrate/nitrite levels [58]. Nitrite levels in the urine were significantly decreased by L-Arg and D-Arg in comparison with the control (Fig 5), which could possibly be due to the diversion of the L-Arg to the arginase urea pathway at higher doses [62, 78] and arginase I protein expression was significantly elevated in the liver (Fig 3). For the D-Arg group, a possible explanation could be competition with endogenous L-Arg resulting in reduced NO production (Fig 5). In the upper small intestine, nitrite levels were significantly reduced by oral D-Arg compared to L-Arg group and control (Fig 5), which again questions the relevance of this finding. The effects of D-Arg cannot be explained due to lack of studies with D-Arg on its metabolic pathways and on its physiological impact with chronic

administration. In high salt diet-induced hypertensive rats administration of oral L-Arg at 100 mg/kg/d for 12 wks prevented the reduced eNOS mRNA expression in the aorta and increased NO levels [72].

The other product of L-Arg through NOS is L-citrulline [6]. It has been shown that part of NO produced is from L-citrulline [79] and L-citrulline might be a better supplement for L-Arg [80] with therapeutic potential [81]. We also performed a citrulline assay and did not find significant changes in L-citrulline levels (Table 4). This result aligns with our previous study where oral L-Arg and D-Arg did not affect L-citrulline levels in the plasma and select organs [2]. In streptozotocin-induced diabetic mice, L-Arg administration at 1.5% in drinking water for 9 wks, did not affect plasma L-citrulline levels [74]. Literature reports of the effect of L-Arg supplements on L-citrulline levels are hard to find.

The AGAT pathway produces creatine and plays a role in regulating energy balance [5]. Both oral L-Arg and D-Arg significantly reduced expression of AGAT protein in the liver, compared to control (Fig 6) as seen with our previous study with 500 mg/kg/d [2]. In the kidney, oral L-Arg, but not D-Arg, significantly increased AGAT protein expression compared to both the control and D-Arg groups (Fig 6), which is opposite to our results in the previous study [2], where L-Arg significantly decreased, and D-Arg increased AGAT protein expression in the kidney compared to the control. In a study on rats the administration of L-Arg at 1.8 or 3.6 g/kg/d for 13 wks significantly increased AGAT activity in the kidney at both doses, without affecting plasma creatinine levels [23]. The opposite effect of L-Arg on kidney AGAT protein expression at 500 mg/kg/d for 4 wks and at 1000 mg/kg/d for 16 wks may be due to different doses and durations of treatment. AGAT deficiency in humans is an autosomal recessive inborn error of creatine synthesis which can cause intellectual developmental disability or delay along with myopathy [82], therefore it is important to know if any dose/treatment duration related effect is produced by L-Arg supplements on creatine synthesis and muscle function.

We measured the levels of creatinine, a metabolite of creatine, in the plasma, urine, and various organs. In the urine, both L- and D-Arg at 1000 mg/kg/d for 16 wks led to significantly reduced creatinine levels, whereas in the upper small intestine, the levels were elevated with just oral D-Arg (Table 5). The levels of creatinine in the plasma, liver and kidney were not affected by oral L- or D-Arg (Table 5). Decreased urinary creatinine levels cannot be explained by decreased AGAT protein expression in the liver by both L- and D-Arg (Fig 6). AGAT activity is highest in the kidney where it synthesizes guanidino acetate from L-Arg [12]. The guanidino acetate is then methylated by the liver to form creatine. AGAT protein expression was increased by L-Arg in the kidney (Fig 6). Since urinary creatinine is a reflection of whole body creatine production, and the renal production accounts for 20 percent of it, the decreased urinary creatine is hard to explain in the face of a balanced diet, unless the liver is the primary source of creatine production in rats as proposed by Brosnan and Brosnan [12]. Again, we could not find supporting literature for any effect of D-Arg on creatine in the small intestine or other organs. In our previous study [2], L-Arg at 500 mg/kg/d for 4 wks significantly decreased creatinine levels in the skeletal muscle, and D-Arg increased it in the liver, both compared to control.

The enzyme ADC converts arginine into agmatine, which is then metabolized by agmatinase into putrescine and formation of other polyamines [4, 5]. Polyamines play important roles in cell growth and support for embryonic development [11]. Another source of polyamines is ornithine, a metabolite of the arginase pathway, which acts as a precursor for polyamines [4]. Ornithine decarboxylase (ODC) is the rate limiting enzyme that converts ornithine into polyamines [11]. Thus, it is generally believed that arginase is the primary enzyme and ADC is the alternative enzyme for polyamine synthesis. Both oral L-Arg and

D-Arg significantly increased ADC protein expression in the liver, and significantly decreased it in the upper small intestine, both in comparison to the control (Fig 7A). Increased ADC protein expression in the liver with L-Arg supplementation could have been caused by the upregulated arginase I expression (Fig 3) which may have lowered L-Arg availability for ADC causing an adaptive increase in expression. As well, reduced expression of arginase II protein in the upper small intestine with L-Arg treatment (Fig 3) could have led to decreased metabolism of L-Arg there, subsequently increasing ADC protein expression in the liver in response to high levels of oral L-Arg in the liver. In our previous study [2] oral L-Arg at a lower dose significantly increased ADC protein expression in the liver, which aligns with the current result in the liver with L-Arg. It is hard to understand the reduced expression of both arginase II protein and ADC in the upper small intestine (Fig 3) by oral L-Arg, which cannot be explained by citing reduced arginase II as a cause for an increased expression of ADC protein to handle excess L-Arg. Unique regulation of each enzyme might be responsible for the observed results. We could not find literature reports on the effects of L-Arg supplements on ADC or agmatinase expression or activity.

Agmatinase protein expression and total polyamines levels were measured. Agmatinase breaks down agmatine as a part of the ADC pathway to form polyamines [11]. In the kidney and upper small intestine, oral D-Arg increased agmatinase protein expression, compared to the control and/or L-Arg groups, and decreased it in the brain, compared to the control (Fig 7B). The increased expression may be explained as an adaptive response, with D-Arg supplements suppressing L-Arg from being metabolized by ADC to produce agmatine, or D-Arg blocking agmatine from being metabolized by agmatinase to cause its upregulation. As D-Arg in the body supposedly does not participate in any metabolic pathways, the reasoning behind these results remains in question for now. In our previous study [2] oral D-Arg at a lower dose decreased agmatinase protein expression in the liver compared to control. The HPLC protocol for agmatine measurement [83, 84] could not be reproduced. The levels of total polyamines were significantly increased in the plasma by oral L-Arg in comparison with the control (Table 6). This could be due to increased expression of arginase I in the liver with both oral L-Arg and D-Arg supplementation (Fig 3), which also contributes to polyamine production through the L-ornithine/OAD/putrescine pathway [85]. Referring back to our lower dose arginine study [2], oral D-Arg treatment at 500 mg/kg/d for 4 wks, significantly increased total polyamines levels in the plasma compared to the control, and L-Arg also significantly increased it in the liver compared to the control and D-Arg groups. Polyamines have been shown to inhibit the activity of ADC [86]. ADC and AGAT enzyme activities could not be measured as we could not find assay kits.

One major limitation of our study is the oral administration in drinking water rather than a bolus oral dose, the way human subjects would take it. This might have impacted the outcomes. For example, the dose in drinking water would enter the body in smaller amounts multiple times a day whenever the rat drinks it. This might not cause a large peak in the plasma concentration, and in fact might be metabolized in the intestines and liver, before it even reaches the circulation. In contrast, when given as a bolus, once or twice a day, a larger amount enters the intestines and the liver, and possibly even the circulation if it is not totally metabolized in the first pass. Another obvious limitation is the caution required to extrapolate these results to humans. We have provided human equivalent doses for rat doses to at least understand doses normally taken by human subjects and what effect equivalent doses have in rats.

As for the practical implications of this study and our previous study [2], it has to be noted that L-arginine is used as a supplement for several conditions, even by healthy people and athletes, in most cases with controversy and lack of knowledge of its overall physiological impact. *There is no mimimum or maximum daily recommended dose.* L-arginine supplements are

taken in a wide dose range from 1.5 to 24 g/day, or approx. 21 to 342 mg/kg/day for a 70 kg adult [18, 19, 76, 87, 88], for a variety of conditions [89] including improvement in aerobic exercise endurance [76, 90] and endothelium-dependent vasodilation in humans with hyper-cholesterolemia.[19, 91]. It is important to remember that high concentrations of L-arginine can also produce toxicity as seen in hyperargininemia, a rare urea cycle defect due to arginase deficiency, in which there is impaired conversion of arginine to urea and ornithine, and CNS pathology [92]. Thus, there is a great need for a safe and effective dose range and awareness among users about the potential adverse effects on one or many of the enzymes and their metabolites of the L-arginine metabolic pathways. An excess of L-Arg supplement intake may be diverted to arginase and upregulate it and cause NO deficiency [62]. The results of this study and our previous study [2], and our future planned studies with different doses for different durations will hopefully contribute form an evidence based dose range that and any possible associated adverse effects.

As for future research directions, we will be conducting studies in rats with a dose of 500 mg/kg/d for 16 wks and 1000 mg/kg/d for 4 wks as a part of our overall project on arginine supplements. In addition, molecular studies with both L-Arg and D-Arg, especially the latter, would significantly add to knowledge about the mechanisms of these substrates on their enzymes and their regulation. It would clarify if indeed D-Arg is an active isomer on one or more of the enzymes that use L-Arg. It can also help to clarify if D-Arg can inhibit any enzyme from using L-Arg, even if D-Arg itself is inactive. There is also a lack of pharmacokinetic data of L-Arg and D-Arg at different doses administered by oral gavage, in drinking water and intravenously as a bolus or as an infusion. In every study using oral L-Arg supplements, animal or clinical, arginase expression/activity or ornithine/urea levels should always be measured to determine if supplemental L-Arg is being diverted to the arginase metabolic pathway at the detriment of NOS pathway. Arginase protein expression/activity are upregulated in many types of cancers, possibly through polyamine-mediated effect on cell proliferation and growth [65], which also necessitates noting the effect of L-Arg supplements on arginase and polyamines. Similarly, the effect of L-Arg supplements on AGAT and creatine synthesis [82] should also be studied when they are used in clinical studies.

The important question is in which conditions would L-Arg supplements be useful? The most popular use and promotion of use for its NO forming substrate. Its use to increase NO production by healthy people, including athletes, is questionable, simply because there is more L-Arg than is needed by eNOS [39]. L-Arg supplements may prove beneficial in conditions where a deficiency of NO is demonstrable, e.g. increased superoxide and oxidative stress [18–20], increased ADMA levels or decreased L-Arg/ADMA ratio, which are commonly encountered in some forms of hypertension, diabetes, atherosclerosis etc. [44, 45]; or in non-cancerous conditions where there is upregulation of arginase [65], including sepsis where there are reduced L-Arg levels and excessive protein catabolism [93]. L-Arg supplementation has been useful to treat urea cycle defects [94], however high L-Arg doses may prove toxic for the brain if there is deficiency of for example, guanidinoacetate methyltransferase [95] in the AGAT-creatine synthetic pathway. L-Arg supplements at doses of 0.15 to 0.3 g/kg/d for 18 months have been useful to prevent mitochondrial encephalopathy, lactic acidosis, and stroke-like episodes (MELAS) [96]. L-Arg supplements may prove useful for wound healing [68]. The most important point in the clinical usefulness of L-Arg supplements is to determine a consistent effective dose and duration for a specific condition, and it may turn out to be different for different conditions. It may also turn out that intermittent administration with say one week on and one week off may also be more effective, It is quite clear that many more studies are required to safely and effectively use this complicated amino acid.

## Conclusions

In conclusion, both oral L-Arg and D-Arg supplements significantly alter enzyme expression and metabolite levels of L-Arg metabolic pathways, although not in a dose-dependent and treatment duration manner as shown by our current and previous study [2]. This does not support our hypothesis that L-Arg and D-Arg at a higher dose for a longer duration will replicate and magnify their effects in altering the expression/activity of L-Arg metabolizing enzymes and levels of its metabolites, produced by a lower dose for a shorter duration [2]. The results of the current study and the previous study [2] highlights the complex and dynamic regulation of the multiple arginine metabolic pathways and their interactions, and the different response of enzymes to different doses and durations. Our planned studies with 500 mg/kg/d for 16 wks and 1000 mg/kg/d for 4 wks will hopefully clarify these matters. These results underscore the need for caution before taking oral L-Arg supplements, especially for healthy people. At the same time, it must be emphasized that our research has been conducted in young growing rats and the extrapolation of these results to humans taking L-Arg supplements at various ages of their lives and possibly with any disease condition, has to be done with caution, and needs further support from studies in humans.

## Supporting information

**S1 Raw images.**
(PDF)

## Author Contributions

**Conceptualization:** Kaushik Desai.

**Formal analysis:** Dain (Raina) Kim, Sarah Martin, Kaushik Desai.

**Funding acquisition:** Kaushik Desai.

**Investigation:** Dain (Raina) Kim, Sarah Martin.

**Methodology:** Dain (Raina) Kim, Sarah Martin.

**Supervision:** Kaushik Desai.

**Writing – original draft:** Dain (Raina) Kim.

**Writing – review & editing:** Kaushik Desai.

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
