## [Decision Letter · Decision Letter 0]

3 May 2023

PONE-D-23-10457The effects of a comparatively higher dose of 1000 mg/kg/d of oral L- or D-arginine on the L-arginine metabolic pathways in male Sprague-Dawley rats.PLOS ONE

Dear Dr. Desai,

Thank you for submitting your manuscript to PLOS ONE. After careful consideration, we feel that it has merit but does not fully meet PLOS ONE’s publication criteria as it currently stands. Therefore, we invite you to submit a revised version of the manuscript that addresses the points raised during the review process.

We look forward to receiving your revised manuscript.

Kind regards,

Kanhaiya Singh, Ph.D

Academic Editor

PLOS ONE

Journal Requirements:

   "This research work was funded by the Alexander Graham Bell Canada Graduate Scholarship from Natural Sciences and Engineering Council of Canada (NSERC) to Dain Kim and a Discovery grant from NSERC to Kaushik Desai."

   "KD- Grant # RGPIN-2016-03951, Natural Sciences and Engineering Council of Canada (NSERC), Discovery Grant, https://www.nserc-crsng.gc.ca/index_eng.asp. The funder had no role in study design, data collection and analysis, decision to publish, or preparation of the manuscript.

DK - Natural Sciences and Engineering Council of Canada (NSERC), https://www.nserc-crsng.gc.ca/index_eng.asp., Alexander Graham Bell Canada Graduate Scholarship, The funder had no role in study design, data collection and analysis, decision to publish, or preparation of the manuscript."

Additional Editor Comments:

Although the reviewers have found this study of interest, they have recommended to elaborate the results section about the logical flow of information.

Reviewers' comments:

Reviewer's Responses to Questions

**Comments to the Author**

1. Is the manuscript technically sound, and do the data support the conclusions?

Reviewer #1: Yes

Reviewer #2: Partly

2. Has the statistical analysis been performed appropriately and rigorously? 

Reviewer #1: Yes

Reviewer #2: Yes

3. Have the authors made all data underlying the findings in their manuscript fully available?

Reviewer #1: Yes

Reviewer #2: No

4. Is the manuscript presented in an intelligible fashion and written in standard English?

Reviewer #1: Yes

Reviewer #2: Yes

5. Review Comments to the Author

Reviewer #1: In this manuscript, the authors are testing the effects of long-duration use of oral L-arginine and D-arginine in nine week old Sprague-Dawley rats. The authors measured the activity of different enzymes such as L-arginine metabolizing enzymes, and also tested the levels of their metabolites in various organs and in the plasma. The results showed that long-duration use of oral L-arginine and D-arginine altered the levels of different metabolites which was not in conjunction with their hypothesis. Thus, the authors specify the dynamic regulation of multiple L-arginine metabolic pathways and propose further studies with different doses and different duration to better assess the use of oral L-arginine.

Overall, the manuscript is well written and the results support the conclusions. I have a few minor points that I have highlighted below:

1. In the materials and methods, for most of the sections, the authors write the reasoning and logic behind the methods. I would recommend to mention just the methods in this section. The explanation, logic, reasoning can be added in the results section of the manuscript.

2. Line 93, male is mentioned twice.

3. Instead of using % signs in text, please use the word percent.

4. Line 179, instead of h, please write hour.

5. In the figures, for graphs, please use asterisks for significance, instead of writing the p value.

Reviewer #2: The manuscript provides an informative presentation of the study's results and discussions on the effects of oral L-Arginine (L-Arg) and D-Arginine (D-Arg) supplementation. It covers various aspects, including the lack of effect on mean arterial pressure (MAP) and heart rate, the potential benefits of L-Arg supplements in disease conditions, the complexities of L-Arg metabolism and nitric oxide (NO) production, and the impact on arginase, nitric oxide synthase (NOS), and the arginine-glycine amidinotransferase (AGAT) pathway.

However, there are areas that could be improved. The interpretation of the results in the discussion section would benefit from further elaboration. For example, if there was no significant effect on MAP and heart rate, exploring potential reasons and discussing influencing factors or variables would enhance the interpretation. Addressing limitations in the study design and their potential impact on the outcomes would also be valuable.

Additionally, the discussion could include practical implications of the findings and suggest future research directions. Exploring the application of the results in clinical settings or relevant fields, identifying specific populations or conditions that might benefit from L-Arg supplementation, and proposing avenues for future studies based on the limitations or unanswered questions from the current study would provide valuable insights.

Comparative analysis with previous research or existing literature is lacking in the discussion. While previous studies on L-Arg supplementation are briefly mentioned, a comprehensive analysis comparing the current findings to those studies would enhance the understanding of the significance and contextualize the results within the broader scientific landscape.

In summary, the manuscript would benefit from a more critical analysis, interpretation, and contextualization of the findings. Providing a comprehensive understanding of the implications and significance of the observed effects of L-Arg and D-Arg supplementation on the various parameters studied, along with comparing the results to previous research, would improve the overall quality of the discussion.

6. PLOS authors have the option to publish the peer review history of their article (what does this mean?). If published, this will include your full peer review and any attached files.

Reviewer #1: No

Reviewer #2: No

---

## [Author Response · Author response to Decision Letter 0]

25 Jun 2023

PONE-D-23-10457

The effects of a comparatively higher dose of 1000 mg/kg/d of oral L- or D-arginine on the L-arginine metabolic pathways in male Sprague-Dawley rats.

PLOS ONE

Editor’s comments:

Reply: We have ensured that our manuscript meets PLOS ONE’s style requirements.

Please remove any funding-related text from the manuscript and let us know how you would like to update your Funding Statement.

Reply: We have removed funding statement from the manuscript text.

 Currently, your Funding Statement reads as follows: 

 "KD- Grant # RGPIN-2016-03951, Natural Sciences and Engineering Council of Canada (NSERC), Discovery Grant, https://www.nserc-crsng.gc.ca/index_eng.asp. The funder had no role in study design, data collection and analysis, decision to publish, or preparation of the manuscript.

DK - Natural Sciences and Engineering Council of Canada (NSERC), https://www.nserc-crsng.gc.ca/index_eng.asp., Alexander Graham Bell Canada Graduate Scholarship, The funder had no role in study design, data collection and analysis, decision to publish, or preparation of the manuscript."

Reply: The above two funding statements are correct.

3. In your Data Availability statement, you have not specified where the minimal data set underlying the results described in your manuscript can be found. 

"Upon re-submitting your revised manuscript, please upload your study’s minimal underlying data set as either Supporting Information files or to a stable, public repository and include the relevant URLs, DOIs, or accession numbers within your revised cover letter. 

Reply: We have uploaded our mimimal underlying data set on DRYAD: https://datadryad.org/stash/share/4jH7mMbb8DzurfoCJzMG1A1pjAyirfB1hJPW9AzPCME DOI https://doi.org/10.5061/dryad.x95x69pqm

Additional Editor Comments:

Although the reviewers have found this study of interest, they have recommended to elaborate the results section about the logical flow of information.

Reply: We have elaborated the results section and the discussion as suggested by the reviewers.

Reviewers' comments:

Reviewer #1: In this manuscript, the authors are testing the effects of long-duration use of oral L-arginine and D-arginine in nine week old Sprague-Dawley rats. The authors measured the activity of different enzymes such as L-arginine metabolizing enzymes, and also tested the levels of their metabolites in various organs and in the plasma. The results showed that long-duration use of oral L-arginine and D-arginine altered the levels of different metabolites which was not in conjunction with their hypothesis. Thus, the authors specify the dynamic regulation of multiple L-arginine metabolic pathways and propose further studies with different doses and different duration to better assess the use of oral L-arginine.

Reply: We are grateful to the reviewer for allocating their time to review our manuscript and provide valuable suggestions to improve it.

Overall, the manuscript is well written and the results support the conclusions. I have a few minor points that I have highlighted below:

1. In the materials and methods, for most of the sections, the authors write the reasoning and logic behind the methods. I would recommend to mention just the methods in this section. The explanation, logic, reasoning can be added in the results section of the manuscript.

Reply: As suggested by the reviewer, paragraphs with the reasoning and logic behind methods have been moved to appropriate results sections. One paragraph about the dose of arginine used and its relevance has been moved to discussion since there was no appropriate results section for it. We hope the reviewer will find it reasonable. 

2. Line 93, male is mentioned twice.

Reply: Thank you for pointing out this error. It has been corrected.

3. Instead of using % signs in text, please use the word percent.

Reply: This has been corrected throughout the text.

4. Line 179, instead of h, please write hour.

Reply: “h” has been replaced with “hour” throughout the text.

5. In the figures, for graphs, please use asterisks for significance, instead of writing the p value.

Reply: We have inserted asterisks instead of p value in all the figures. 

We hope the reviewer will find our corrections satisfactory.

Reviewer #2: The manuscript provides an informative presentation of the study's results and discussions on the effects of oral L-Arginine (L-Arg) and D-Arginine (D-Arg) supplementation. It covers various aspects, including the lack of effect on mean arterial pressure (MAP) and heart rate, the potential benefits of L-Arg supplements in disease conditions, the complexities of L-Arg metabolism and nitric oxide (NO) production, and the impact on arginase, nitric oxide synthase (NOS), and the arginine-glycine amidinotransferase (AGAT) pathway.

Reply: We grateful to the reviewer for allocating their time to review our manuscript and providing valuable suggestions to improve it. We truly appreciate the encouraging comments. 

However, there are areas that could be improved. The interpretation of the results in the discussion section would benefit from further elaboration. For example, if there was no significant effect on MAP and heart rate, exploring potential reasons and discussing influencing factors or variables would enhance the interpretation. Addressing limitations in the study design and their potential impact on the outcomes would also be valuable.

Reply: As suggested by the reviewer we have now discussed possible reasons and influencing factors for the lack of effect on MAP and heart rate. We have also addressed limitations in study design and their potential impact on the outcomes. We notice that these suggestions have improved our discussion.

Additionally, the discussion could include practical implications of the findings and suggest future research directions. Exploring the application of the results in clinical settings or relevant fields, identifying specific populations or conditions that might benefit from L-Arg supplementation, and proposing avenues for future studies based on the limitations or unanswered questions from the current study would provide valuable insights.

Reply: We have included paragraphs discussing the practical implications of our findings and suggested future research directions which can add significantly to the knowledge database of L-Arg supplements. We have suggested populations that could benefit from oral L-Arg supplements.

Comparative analysis with previous research or existing literature is lacking in the discussion. While previous studies on L-Arg supplementation are briefly mentioned, a comprehensive analysis comparing the current findings to those studies would enhance the understanding of the significance and contextualize the results within the broader scientific landscape.

Reply: We have compared our current results with our previous study in greater details now and also provided relevant findings reported in the literature, wherever available. Based on these comparisons we have tried provide the significance of the results as it relates to the use of L-Arg supplements in a broad and varied scientific field.

In summary, the manuscript would benefit from a more critical analysis, interpretation, and contextualization of the findings. Providing a comprehensive understanding of the implications and significance of the observed effects of L-Arg and D-Arg supplementation on the various parameters studied, along with comparing the results to previous research, would improve the overall quality of the discussion.

Reply: After implementing the suggestions we notice a definite improvement in the manuscript. We hope the reviewer will find our corrections satisfactory.

---

## [Decision Letter · Decision Letter 1]

20 Jul 2023

The effects of a comparatively higher dose of 1000 mg/kg/d of oral L- or D-arginine on the L-arginine metabolic pathways in male Sprague-Dawley rats.

PONE-D-23-10457R1

Dear Dr. Desai,

We’re pleased to inform you that your manuscript has been judged scientifically suitable for publication and will be formally accepted for publication once it meets all outstanding technical requirements.

Kind regards,

Kanhaiya Singh, Ph.D

Academic Editor

PLOS ONE

Additional Editor Comments (optional):

Reviewers' comments:

Reviewer's Responses to Questions

**Comments to the Author**

1. If the authors have adequately addressed your comments raised in a previous round of review and you feel that this manuscript is now acceptable for publication, you may indicate that here to bypass the “Comments to the Author” section, enter your conflict of interest statement in the “Confidential to Editor” section, and submit your "Accept" recommendation.

Reviewer #1: All comments have been addressed

2. Is the manuscript technically sound, and do the data support the conclusions?

Reviewer #1: Yes

3. Has the statistical analysis been performed appropriately and rigorously? 

Reviewer #1: Yes

4. Have the authors made all data underlying the findings in their manuscript fully available?

Reviewer #1: Yes

5. Is the manuscript presented in an intelligible fashion and written in standard English?

Reviewer #1: Yes

6. Review Comments to the Author

Reviewer #1: (No Response)

7. PLOS authors have the option to publish the peer review history of their article (what does this mean?). If published, this will include your full peer review and any attached files.

Reviewer #1: No

---

## [Editor Report · Acceptance letter]

24 Jul 2023

PONE-D-23-10457R1 

The effects of a comparatively higher dose of 1000 mg/kg/d of oral L- or D-arginine on the L-arginine metabolic pathways in male Sprague-Dawley rats 

Dear Dr. Desai:

I'm pleased to inform you that your manuscript has been deemed suitable for publication in PLOS ONE. Congratulations! Your manuscript is now with our production department. 

Kind regards, 

on behalf of

Dr. Kanhaiya Singh 

Academic Editor

PLOS ONE